# Improving Multi-step RAG with Hypergraph-based Memory for Long-Context Complex Relational Modeling

## Abstract

Multi-step retrieval-augmented generation (RAG) has become a widely adopted strategy for enhancing large language models (LLMs) on tasks that demand global comprehension and intensive reasoning. Many RAG systems incorporate a working memory module to consolidate retrieved information. However, existing memory designs function primarily as passive storage that accumulates isolated facts for the purpose of condensing the lengthy inputs and generating new sub-queries through deduction. This static nature overlooks the crucial high-order correlations among primitive facts, the compositions of which can often provide stronger guidance for subsequent steps. Therefore, their representational strength and impact on multi-step reasoning and knowledge evolution are limited, resulting in fragmented reasoning and weak global sense-making capacity in extended contexts.

We introduce HGMEM, a hypergraph-based memory mechanism that extends the concept of memory beyond simple storage into a dynamic, expressive structure for complex reasoning and global understanding. In our approach, memory is represented as a hypergraph whose hyperedges correspond to distinct memory units, enabling the progressive formation of higher-order interactions within memory. This mechanism connects facts and thoughts around the focal problem, evolving into an integrated and situated knowledge structure that provides strong propositions for deeper reasoning in subsequent steps. We evaluate HGMEM on several challenging datasets designed for global sense-making. Extensive experiments and in-depth analyses show that our method consistently improves multi-step RAG and substantially outperforms strong baseline systems across diverse tasks.

## 1 Introduction

Single-step retrieval-augmented generation (RAG) often proves insufficient for resolving complex queries within long contexts (Trivedi et al., 2023; Shao et al., 2023; Cheng et al., 2025), motivating the shift toward multi-step RAG methods that iteratively interleave retrieval with reasoning. To effectively capture dependencies across steps and condense the lengthy processing history, many approaches incorporate working memory mechanisms inspired by human cognition (Lee et al., 2024; Zhong et al., 2024). However, current memory-enhanced multi-step RAG methods still face challenges in complex relational modeling, especially for resolving global sense-making tasks over long contexts.

During multi-step RAG execution, a straightforward implementation of working memory mechanism is to let a large language model (LLM) summarize the interaction history into a plaintext description of current problem-solving state. This strategy has been widely adopted since early studies (Li et al., 2023; Trivedi et al., 2023) as well as in commercial systems (Jones, 2025; Shen & Yang, 2025). Nonetheless, such unstructured memory mechanisms cannot be manipulated with sufficient accuracy across steps and often lose the ability to back-trace references to retrieved texts. Consequently, recent research has shifted toward structured or semi-structured working memory, typically with predefined schemas such as relational tables (Lu et al., 2023), knowledge graphs (Oguz et al., 2022; Xu et al., 2025), or event-centric bullet points (Wang et al., 2025).

However, existing memory mechanisms often treat memory as static storage that continually accumulates meaningful but primitive facts. This view overlooks the evolving nature of human working memory, which incrementally incorporates higher-order correlations from previously memorized content. This capacity is particularly crucial for resolving global sense-making tasks that involve complex relational modeling over long contexts. In such scenarios, the required knowledge for tackling a query is often composed of complex structures that extend beyond predefined schemas, and reasoning over long lists of primitive facts is both inefficient and prone to confusion with mixed or irrelevant information. Current memory mechanisms in multi-step RAG systems lack these abilities, preventing memory from effectively guiding LLMs' interaction with external data sources. These limitations highlight the need for a working memory with stronger representational capacity.

In this paper, we propose a hypergraph-based memory mechanism (HGMem) for multi-step RAG systems, which enables memory to evolve into more expressive structures that support complex relational modeling to enhance LLMs' understanding over long contexts. Hypergraphs, as a generalization of graphs, are particularly well-suited for this purpose (Feng et al., 2019). In our design, memory is structured as a hypergraph composed of hyperedges, each treated as a distinct memory point that represents a specific perspective of the memorized information. Initially, these memory points encode low-order primitive facts. As the LLM interacts with external environments, higher-order correlations among memory points gradually emerge and are progressively integrated into the memory through update, insertion, and merging operations. At each step before response generation, the LLM examines the current memory and generates subqueries, enabling adaptive memory-based evidene retrieval for both focused local investigation and broad global exploration.

This rich and structured memory facilitates broader contextual awareness and stronger reasoning in real-world applications by offering several advantages. First, it maintains an **integrated body of knowledge** around the focal problem by synthesizing primitive evidence and intermediate thoughts, typically going *beyond predefined schemas* and providing a *global perspective* over the evidence. Second, it offers **structured and accurate guidance** for the LLM's sustained interactions in two ways: (1) enabling subsequent reasoning to start from representational propositions rather than from a long list of disparate primitive facts; and (2) leveraging the topological structure of hypergraph to guide subquery generation and evidence retrieval in a more accurate manner.

We conduct extensive experiments on several challenging tasks involving global sense-making questions within long contexts. The results show that our HGMem achieves significant improvements over competitive RAG baselines, confirming the advantages.

## 2 RELATED WORK

### 2.1 WORKING MEMORY MECHANISMS FOR MULTI-STEP RAG

Starting from ReAct (Yao et al., 2023), many multi-step RAG systems have incorporated reflections to integrate available information for subsequent decisions. These reflections can be regarded as a simple form of memory. With the development of structured indexing for RAG, working memory also borrows this idea. Prevailing studies (Li et al., 2023; 2025a; Shen & Yang, 2025; Chhikara et al., 2025; Xu et al., 2025) save agent behavior, such as task decomposing, execution tracking, and result verification, to manage task context more effectively, representing a step toward explicit working memory for complex multi-agent coordination. This idea also matured in chain-of-thought (CoT) and multi-round RAG, where working memory is represented as iteratively updated records of reasoning steps or retrieved evidence. For example, IRCOT (Trivedi et al., 2023) and ComoRAG (Wang et al., 2025) employ a dynamic memory workspace to iteratively consolidate past knowledge or steps and incorporate new evidence, supporting scalable and iterative reasoning across multiple steps.

Some studies take a step further to adopt a graph-structured working memory to enhance multi-step RAG (Liu et al., 2024; Li et al., 2025a). ERA-CoT (Liu et al., 2024) aids LLMs in understanding context through a series of pre-defined reasoning substeps performing entity-relationship analysis. KnowTrace (Li et al., 2025a) equips LLMs with a graph-based working memory to trace relevant knowledge through multi-step RAG execution. However, the working memories of these graph-enhanced work do not effectively support modeling high-order correlations among multiple entities/relationships as each edge in their graphs can intrinsically describe at most binary relationships. By contrast, due to the high-order nature of hypergraph structure, our HGMem naturally enables

its working memory to evolve into more expressive forms capable of flexibly modeling high-order $n$-ary ($n > 2$) relations. This advantage helps to fully unleash the reasoning capability of LLMs for multi-step RAG, especially crucial for resolving global sense-making questions that require complex reasoning and deep understanding over long contexts.

## 2.2 RAG WITH STRUCTURED KNOWLEDGE INDEX

There is a long line of work that studies managing extended corpora through structured knowledge indexing to enhance RAG. Though different from our focus on working memory mechanism, these work can be viewed as building structured (and static) long-term memory before actually tackling user queries, thus are relevant. Specifically, tree-structured methods, such as RAPTOR (Sarthi et al., 2024), T-RAG (Fatehkia et al., 2024), and TreeRAG (Tao et al., 2025), organize text chunks or entity hierarchies, enabling multi-level or bidirectional retrieval to enhance context integration. Another line of research focuses on building graph-structured index to flexibly represent knowledge for enhancing RAG systems (Xu et al., 2024a; Edge et al., 2024; Guo et al., 2024; Li et al., 2025b). For example, GraphRAG (Edge et al., 2024) and LightRAG (Guo et al., 2024) build entity graphs and community-level summaries, or leverage graph-enhanced indexing for dual-level retrieval, leading to improvements in global reasoning, retrieval efficiency, and response diversity. CAM (Li et al., 2025b) proposes a constructivist agentic memory that flexibly assimilates and accommodates input texts within a hierarchical graph. HypergraphRAG (Luo et al., 2025) and PropRAG (Wang, 2025) adopt hypergraph to build their structured knowledge index and design retrieval/search algorithms for query resolving. In addition, there are also a range of other memory mechanisms, essentially structured knowledge index, that simulate long contexts or dialog histories as long-term memory to improve RAG systems. According to the form of memory representation, they can be basically classified as contextual memory (Chen et al., 2023; Gutierrez et al., 2024; Lee et al., 2024; Li et al., 2024b; Gutiérrez et al., 2025) and parametric memory (Qian et al., 2025).

However, these existing studies merely leverage their structured index (or memory) as static storage, which are typically constructed during an offline indexing stage before actually responding to user queries.

## 3 METHODOLOGY

We introduce HGMEM, the hypergraph-based memory mechanism designed to facilitate better contextual awareness and reasoning in multi-step RAG settings with structured data sources, especially for long-context tasks that require complex global sense-making.

### 3.1 PROBLEM FORMULATION

In this work, we consider the kind of tasks for LLMs to resolve a query based on a given document. Besides the plain texts, we assume that the document has been preprocessed into a graph through an offline graph-building stage, where entities and relationships are extracted from the document passage. Formally, let us denote the document as $\mathcal{D}$ segmented into a set of small manageable text chunks $\{d_1, d_2, ..., d_{|\mathcal{D}|}\}$, and the derived graph as $\mathcal{G}$ composed of nodes $\mathcal{V}_{\mathcal{G}}$ and edges $\mathcal{E}_{\mathcal{G}}$ corresponding to the extracted entities and relationships, respectively. Each node $v \in \mathcal{V}_{\mathcal{G}}$ or edge $e \in \mathcal{E}_{\mathcal{G}}$ is associated with the source text chunks in which its embodied entity/relationship appears, which is recorded during the offline graph construction. Meanwhile, the nodes, edges, and text chunks are embedded into high-dimensional vectors for vector-based retrieval. For resolving the query, LLMs have access to both the document and its derived graph as structured data sources.

### 3.2 MULTI-STEP RAG SYSTEM WITH MEMORY

When dealing with tasks requiring comprehensive understanding, especially over long context, RAG systems usually resort to multi-step approaches with an underlying memory mechanism, where retrieval operations are interleaved with intermediate reasoning to support broader contextual awareness.

Given a target query $\hat{q}$, the LLM iteratively interacts with $\mathcal{D}$ and $\mathcal{G}$ while managing a memory $\mathcal{M}$ to store relevant information for ultimately resolving $\hat{q}$. During each interaction step $t$, the LLM judges

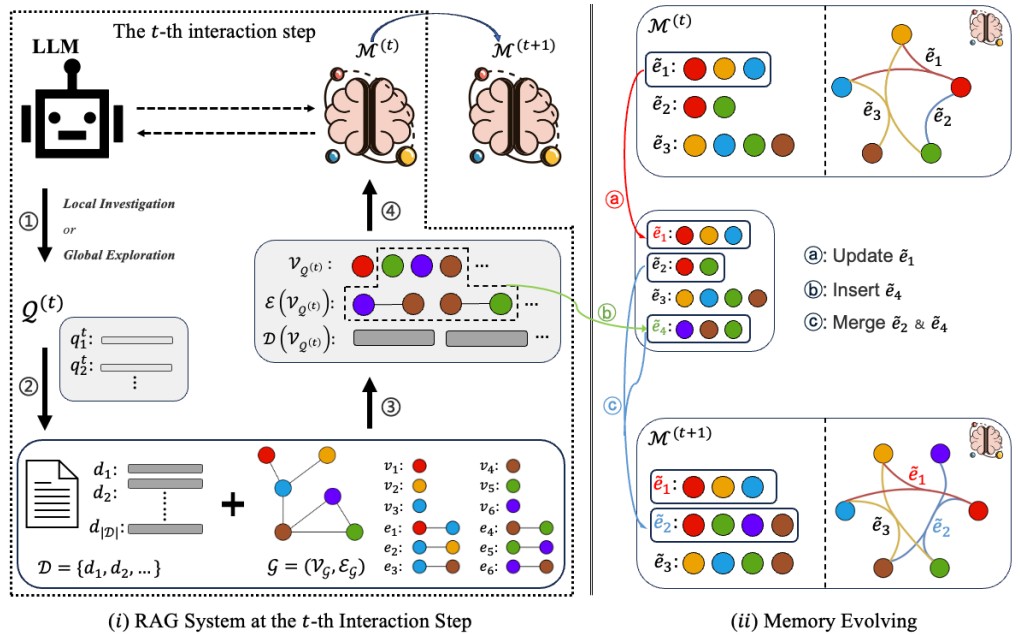

(i) RAG System at the $t$-th Interaction Step      (ii) Memory Evolving

Figure 1: (i) The RAG system at its $t$-th interaction step. ①: The LLM adaptively generates a set of subqueries $\mathcal{Q}^{(t)}$ for either local investigation or global exploration (see Section 3.4). ②: $\mathcal{Q}^{(t)}$ are used to retrieve information from $\mathcal{D}$ and $\mathcal{G}$. ③: $\mathcal{V}_{\mathcal{Q}^{(t)}}$, $\mathcal{E}(\mathcal{V}_{\mathcal{Q}^{(t)}})$ and $\mathcal{D}(\mathcal{V}_{\mathcal{Q}^{(t)}})$ are obtained through graph-based indexing and vector-based matching. ④: The LLM evolves current memory $\mathcal{M}^{(t)}$ into $\mathcal{M}^{(t+1)}$ using Equation 2. (ii) The structure of our proposed hypergraph-based memory that evolves through update, insertion and merging operations.

whether the content of current memory has been sufficient with respect to the target query. If the memory is deemed sufficient, it immediately produces a response. Otherwise, it analyzes current memory and generates several subqueries $\mathcal{Q}^{(t)}$ that aim at fetching more information from external environment to enrich the memory. The prompts for generating subqueries are given in Appendix E.

Let $\mathcal{R}_\mathcal{V}(\mathcal{Q})$ define the entity retrieval operation fetching the most relevant nodes to a query set $\mathcal{Q}$ from a candidate node set $\mathcal{V}$ using vector-based matching:

$$\mathcal{R}_\mathcal{V}(\mathcal{Q}) = \bigcup_{q\in\mathcal{Q}} \underset{v\,\in\,\mathcal{V}}{\operatorname{argmax}^{n_v}}(\operatorname{sim}(\mathbf{h}_q, \mathbf{h}_v)), \tag{1}$$

where $n_v$ is the number of retrieved entities per query, $\mathbf{h}_q$ is the vector representation of $q$, $\mathbf{h}_v$ is the vector representation of $v$, and $\operatorname{sim}(\cdot, \cdot)$ is the cosine similarity function.

As illustrated in Figure 1 (i), at the $t$-th step, if the LLM proceeds to generate subqueries $\mathcal{Q}^{(t)}$ based on current memory $\mathcal{M}^{(t)}$ maintained until the previous step, it retrieves a set of the most relevant entities $\mathcal{V}_{\mathcal{Q}^{(t)}} = \mathcal{R}_{\mathcal{V}_\mathcal{G}}(\mathcal{Q}^{(t)})$ from $\mathcal{V}_\mathcal{G}$. Then, via graph-based indexing, the relationships and text chunks associated with the entities in $\mathcal{V}_{\mathcal{Q}^{(t)}}$ are also obtained, represented as $\mathcal{E}(\mathcal{V}_{\mathcal{Q}^{(t)}})$ and $\mathcal{D}(\mathcal{V}_{\mathcal{Q}^{(t)}})$, respectively.[1] Subsequently, the LLM analyzes and consolidates these retrieved information into the memory, evolving memory into $\mathcal{M}^{(t+1)}$, which can be formalized as

$$\mathcal{M}^{(t+1)} \leftarrow \operatorname{LLM}(\mathcal{M}^{(t)}, \mathcal{V}_{\mathcal{Q}^{(t)}}, \mathcal{E}(\mathcal{V}_{\mathcal{Q}^{(t)}}), \mathcal{D}(\mathcal{V}_{\mathcal{Q}^{(t)}})). \tag{2}$$

Note that, at the initial step ($t=0$), we treat the target query $\hat{q}$ as a special subquery belonging to $\mathcal{Q}^{(0)}$, i.e. $\mathcal{Q}^{(0)}=\{\hat{q}\}$. Further details about the memory storage, subquery generation and the dynamic of memory evolving will be elaborated in Section 3.3, Section 3.4 and Section 3.5, respectively.

---

[1]We also use vector-based filtering to keep at most $n_e$ relationships and $n_d$ text chunks.

### 3.3 HYPERGRAPH-BASED MEMORY STORAGE

When the LLM interacts with the document $\mathcal{D}$ and the graph $\mathcal{G}$, it continuously consolidates relevant information into the memory storage $\mathcal{M}$, which is modeled as a hypergraph:

$$\mathcal{M} = (\mathcal{V}_\mathcal{M}, \tilde{\mathcal{E}}_\mathcal{M}), \tag{3}$$

where $\mathcal{V}_\mathcal{M} = \{v_1, v_2, ...\}$ is the vertex set and $\tilde{\mathcal{E}}_\mathcal{M} = \{\tilde{e}_1, \tilde{e}_2, ...\}$ is the hyperedge set. It should be noted that the vertices in $\mathcal{V}_\mathcal{M}$ are actually equivalent to those nodes in $\mathcal{V}_\mathcal{G}$, both embodying identified entities. Particularly, $\mathcal{V}_\mathcal{M}$ is a subset of $\mathcal{V}_\mathcal{G}$. In our implementation, we ensure that each vertex $v_i \in \mathcal{V}_\mathcal{M}$ must also exist in $\mathcal{G}$.[2] Formally, every vertex $v_i \in \mathcal{V}_\mathcal{M}$ is represented as

$$v_i = (\Omega_{v_i}^{ent}, \mathcal{D}_{v_i}), \tag{4}$$

where $\Omega_{v_i}^{ent}$ stands for the information of its embodied entity including name and description, and $\mathcal{D}_{v_i}$ denotes the set of text chunks associated with this vertex $v_i$. Similarly, every hyperedge $\tilde{e}_j \in \mathcal{E}_\mathcal{M}$ is represented as

$$\tilde{e}_j = (\Omega_{\tilde{e}_j}^{rel}, \mathcal{V}_{\tilde{e}_j}), \tag{5}$$

where $\Omega_{\tilde{e}_j}^{rel}$ characterizes the description of the embodied relationship and $\mathcal{V}_{\tilde{e}_j}$ is the set of involved vertices subordinate to this hyperedge $\tilde{e}_j$. Particularly, the hyperedges can be treated as separate memory points, each of which corresponds to a certain aspect of the entire information stored in current memory, as shown in Figure 1 (*ii*). Unlike those binary edges $\mathcal{E}_\mathcal{G}$ that connect at most two nodes in the external graph, a hyperedge can connect an arbitrary number (two or more) of vertices. In this way, our hypergraph-based memory is capable of flexibly modeling high-order correlation among multiple vertices ($n \geq 2$). As a result, the whole memory as a hypergraph can effectively support complex relational modeling, ensuring strong expressiveness to enhance LLMs' reasoning.

### 3.4 ADAPTIVE MEMORY-BASED EVIDENCE RETRIEVAL

As described in Section 3.2, at each step $t$ of our RAG workflow, with respect to the target query, the LLM determines whether to immediately produce a response or proceed to acquire more information from the external documents $\mathcal{D}$ and graph $\mathcal{G}$. If current memory $\mathcal{M}^{(t)} = (\mathcal{V}_\mathcal{M}^{(t)}, \tilde{\mathcal{E}}_\mathcal{M}^{(t)})$ is deemed insufficient, the LLM first analyzes $\mathcal{M}^{(t)}$ and generates several subqueries $\mathcal{Q}^{(t)}$ indicating what to further explore. Specifically, we design an adaptive memory-based evidence retrieval strategy for either local investigation or global exploration with $\mathcal{Q}^{(t)}$:

(i) Local Investigation: When the LLM plans to more deeply investigate some specific memory points, its generated subqueries are utilized to trigger local evidence retrieval over $\mathcal{G}$. Concretely, suppose a $q \in \mathcal{Q}^{(t)}$ especially targets at inspecting $\tilde{e}_j \in \tilde{\mathcal{E}}_\mathcal{M}^{(t)}$, the nodes corresponding to the vertices $\mathcal{V}_{\tilde{e}_j}$ subordinate to $\tilde{e}_j$ are used as anchor nodes on $\mathcal{G}$. Thereafter, using the operation defined by Equation 1, entity retrieval is conducted within the neighborhood of these anchors, which is formalized as

$$\mathcal{V}_q = \mathcal{R}_{\mathcal{N}(\mathcal{V}_{\tilde{e}_j})}(q), \tag{6}$$

$$\mathcal{N}(\mathcal{V}_{\tilde{e}_j}) = \bigcup_{v \in \mathcal{V}_{\tilde{e}_j}} (\mathcal{N}_{\mathcal{M}^{(t)}}(v) \cup \mathcal{N}_\mathcal{G}(v)),$$

where $\mathcal{N}_{\mathcal{M}^{(t)}}(v)$ represents the neighboring vertices of $v$ over $\mathcal{M}^{(t)}$ and $\mathcal{N}_\mathcal{G}(v)$ represents the neighboring nodes of $v$ over $\mathcal{G}$.

(ii) Global Exploration: When there are unexplored aspects transcending the scope of current memory, the LLM resorts to generating subqueries for exploring broader information from the external documents and graph, not pertinent to any existing memory point. For a $q \in \mathcal{Q}^{(t)}$, the process of entity retrieval can be written as

$$\mathcal{V}_q = \mathcal{R}_{\mathcal{C}(\mathcal{M}^{(t)})}(q), \tag{7}$$

$$\mathcal{C}(\mathcal{M}^{(t)}) = \mathcal{V}_\mathcal{G} - \mathcal{V}_{\mathcal{M}^{(t)}},$$

where $\mathcal{C}(\mathcal{M}^{(t)})$ represents the available scope comprised of all nodes except those already existing in current memory.

---

[2]If any vertex does not exist in $\mathcal{V}_\mathcal{G}$, we forcibly insert it, along with its associated relationships, into $\mathcal{G}$.

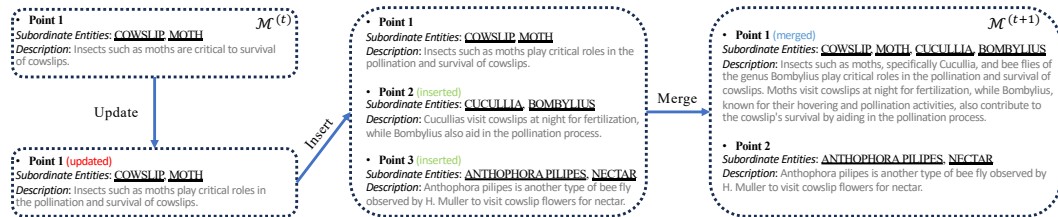

Figure 2: An illustration of memory evolving dynamics. Each point is equivalent to a hyperedge in the hypergraph. $\mathcal{M}^{(t)}$ evolves into $\mathcal{M}^{(t+1)}$ through update, insertion and merging operations.

Then, as in Section 3.2, the associated relationships $\mathcal{E}(\mathcal{V}_q)$ and text chunks $\mathcal{D}(\mathcal{V}_q)$ are obtained via graph-based indexing. Finally, following Equation 2, the LLM evolves its current memory $\mathcal{M}^{(t)}$ into $\mathcal{M}^{(t+1)}$. Under such a strategy, the RAG system is able to adaptively combine both local investigation and global exploration for more flexible information retrieval during interaction with external data sources.

## 3.5 DYNAMIC OF MEMORY EVOLVING

Once a set of subqueries have been generated at the $t$-th step, following Equation 2, the LLM analyzes the retrieved information and consolidates useful content into current memory $\mathcal{M}^{(t)}$, resulting in the evolved memory $\mathcal{M}^{(t+1)}$. As shown in Figure 1 (*ii*), on the basis of hypergraph-based memory storage, the dynamic of memory evolving in our proposed HGMEM involves the following three types of operations:

- *Update*. According to the retrieved information, if there are certain existing memory points whose descriptions should be modified, the update operation will revise the descriptions of corresponding hyperedges without changing their subordinate entities.

- *Insertion*. The insertion operation should be evoked when some content of the retrieved information is suitable to be inserted as additional memory points into the current memory, which creates new hyperedges into the hypergraph.

- *Merging*. After insertion and update, the LLM inspects current memory and selectively merges existing memory points that are more suitable to constitute a single semantically/logically cohesive unit. With respect to the target query $\hat{q}$, suppose the memory points $\tilde{e}_i=(\Omega^{rel}_{\tilde{e}_i}, \mathcal{V}_{\tilde{e}_i})$ and $\tilde{e}_j=(\Omega^{rel}_{\tilde{e}_j}, \mathcal{V}_{\tilde{e}_j})$ are to be merged into a high-order memory point $\tilde{e}_k=(\Omega^{rel}_{\tilde{e}_k}, \mathcal{V}_{\tilde{e}_k})$, its description and subordinate vertices are acquired as

$$\Omega^{rel}_{\tilde{e}_k} \leftarrow \text{LLM}(\Omega^{rel}_{\tilde{e}_i}, \Omega^{rel}_{\tilde{e}_j}, \hat{q}) \tag{8}$$
$$\mathcal{V}_{\tilde{e}_k} = \mathcal{V}_{\tilde{e}_i} \cup \mathcal{V}_{\tilde{e}_j}.$$

Then, the newly merged memory point is added into the hyperedge set $\tilde{\mathcal{E}}_{\mathcal{M}^{(t)}}$ of current memory $\mathcal{M}^{(t)}$. This merging operation over the hypergraph-based memory builds higher-order correlations among multiple existing memory points, facilitating the resolution of queries that require complex relational modeling with disparate facts.

In this way, besides continuously accumulating primitive facts during the LLM's interactions with external data sources, the memory also gradually evolves into more sophisticated forms, capturing higher-order correlations for complex relational modeling. Figure 2 gives a concrete example illustrating the dynamic of memory evolving.

## 3.6 MEMORY-ENHANCED RESPONSE GENERATION

When the LLM exceeds its maximum interaction steps or the content in current memory $\mathcal{M}^{(t)} = (\mathcal{V}^{(t)}_{\mathcal{M}}, \tilde{\mathcal{E}}^{(t)}_{\mathcal{M}})$ has been deemed sufficient, a response is immediately produced according to the information stored in current memory. Concretely, besides the descriptions of all memory points (*i.e.* $\tilde{\mathcal{E}}^{(t)}_{\mathcal{M}}$), the text chunks associated with all the entities $\mathcal{V}^{(t)}_{\mathcal{M}}$ in current memory are also provided to the LLM for producing the final response.

## 4 Experimental Settings

### 4.1 Datasets

We choose generative sense-making question answering (QA) (Edge et al., 2024; Guo et al., 2024) and long narrative understanding (Li et al., 2024a; Xu et al., 2024b; Yu et al., 2025; Kociský et al., 2018; Karpinska et al., 2024; Yen et al., 2025; Zhou et al., 2025) as our evaluation tasks. For generative sense-making QA, similar to the setups used in previous works (Edge et al., 2024; Guo et al., 2024), we retain a portion of long documents with more than 100k tokens from **Longbench V2** (Bai et al., 2025). From each retained document, we use GPT-4o to generate several global sense-making queries that satisfy the following requirements: 1) The queries should target at the overall understanding of the whole provided documents, instead of only concentrating on several specific phrases or sentence pieces. 2) The queries should require high-level understandings and global reasoning over disparate evidence scattered across the whole paragraph. For long narrative understanding, we use three public benchmarks including **NarrativeQA** (Kociský et al., 2018), **NoCha** (Karpinska et al., 2024) and **Prelude** (Yu et al., 2025). Both tasks require global comprehension and complex sense-making over disparate evidence across long contexts. Details about the usage and statistics of data used in our experiments are given in Appendix A.

### 4.2 Implementation Details

**Offline Graph Construction.** For all the datasets used in our experiments, we first segment every document into text chunks of 200 tokens with 50 overlapping tokens between adjacent chunks. Then, GPT-4o is utilized to preprocess each of the chunkized documents into a graph using the open-sourced tool provided by LightRAG (Guo et al., 2024). After building the graph, we adopt *bge-m3* (Chen et al., 2024) as the embedding model to convert all the entities, relationships and text chunks into vector representations managed by *nano vector database*.

**System Deployment and Configuration.** Our RAG system is comprised of the backbone LLM and the hypergraph-based memory. We choose GPT-4o and Qwen2.5-32B-Instruct as the representatives of advanced closed-source and open-source LLMs, respectively. During experiments, GPT-4o is accessed through official API while Qwen2.5-32B-Instruct is locally deployed with VLLM (Kwon et al., 2023). For the configuration of LLM inference, we set the temperature to 0.8 and the maximum number of output tokens to 2,048. As for the hypergraph-based memory, we employ the *hypergraph-db* package to maintain and manage the hypergraph at runtime. The vector representations of the nodes, hyperedges and associated text chunks in the hypergraph are also generated by *bge-m3* embedding model.

### 4.3 Baselines and Evaluation Metrics

In our experiments, we compare our proposed HGMEM to two types of baseline methods, *i.e.* traditional RAG and multi-step RAG, which utilize plain texts and/or graph-structured data sources. Among these methods, DeepRAG (Guan et al., 2025) and ComoRAG (Wang et al., 2025) are equipped with a working memory while the others are not. The details of these comparison methods can be found in Appendix B. To ensure fair comparison, all baselines operate on a similar number of retrieved passages. In the case of single-step RAG, this means retrieving the same average number of text chunks as our HGMEM. For multi-step RAG methods, we approximate comparability by constraining them to rewrite the same maximum number of subqueries and perform the same maximum number of steps, while requiring retrieval of the same average number of chunks per step.

For generative sense-making QA, we adopt the following two metrics to assess the qualities of model responses: 1) **Comprehensiveness** measures how well the model response comprehensively covers and addresses all aspects and necessary details with respect to the target query. 2) **Diversity** indicates how rich and diverse the response is in providing various perspectives and insights related to the query. We employ GPT-4o as the judge to evaluate the model responses according to the grading criteria that gives scores ranging from 0 to 100 based on a two-step scoring scheme, as detailed in Appendix F.

For long narrative understanding, including NarrativeQA, Nocha, and Prelude, we uniformly use prediction accuracy (Acc) as the reported metric. Specifically, for NarrativeQA, prior studies (Bulian et al., 2022; Wang et al., 2024; Zhou et al., 2025) have shown that conventional token-level

Table 1: The overall experimental results on four benchmarks. The second column "**Working Memory**" distinguishes whether the corresponding method is equipped with a working memory that enhances LLMs during RAG execution. The best scores in each dataset are **bolded**. HGMEM consistently outperforms other comparison methods across all datasets.

| Type | Working Memory | Method | Longbench | | NarrativeQA | NoCha | Prelude |
| | | | Comprehensiveness | Diversity | Acc (%) | Acc (%) | Acc (%) |
|---|---|---|---|---|---|---|---|
| *GPT-4o* | | | | | | | |
| Traditional RAG | × | NaiveRAG | 61.62 | 64.20 | 52.00 | 67.46 | 60.00 |
| | × | GraphRAG | 60.39 | 64.02 | 53.00 | 70.63 | 59.26 |
| | × | LightRAG | 61.55 | 63.37 | 44.00 | 71.43 | 61.48 |
| | × | HippoRAG v2 | 58.92 | 61.27 | 34.00 | 72.22 | 54.81 |
| Multi-step RAG | ✓ | DeepRAG | 63.62 | 65.98 | 45.00 | 67.46 | 56.30 |
| | ✓ | ComoRAG | 62.18 | 65.82 | 54.00 | 63.49 | 54.07 |
| Ours | ✓ | HGMEM | **65.73** | **69.74** | **55.00** | **73.81** | **62.96** |
| *Qwen2.5-32B-Instruct* | | | | | | | |
| Traditional RAG | × | NaiveRAG | 61.41 | 62.25 | 37.00 | 64.29 | 52.59 |
| | × | GraphRAG | 60.78 | 62.16 | 44.00 | 62.70 | 50.37 |
| | × | LightRAG | 60.82 | 62.73 | 40.00 | 59.52 | 60.74 |
| | × | HippoRAG v2 | 56.66 | 60.80 | 33.00 | 68.25 | 51.85 |
| Multi-step RAG | ✓ | DeepRAG | 61.45 | 63.56 | 44.00 | 66.40 | 51.11 |
| | ✓ | ComoRAG | 60.74 | 61.28 | 44.00 | 57.60 | 50.37 |
| Ours | ✓ | HGMEM | **64.18** | **66.51** | **51.00** | **70.63** | **62.22** |

metrics such as Exact Match and F1 score usually fail to reflect actual semantic equivalence between hypothesis and reference answer, especially for abstractive answers. Therefore, we also apply GPT-4o for judging whether the LLM's prediction fully entails the reference answer, producing a binary True/False decision.

# 5 RESULTS AND ANALYSIS

## 5.1 OVERALL RESULTS

Table 1 reports the overall results across all evaluation tasks. Our HGMEM consistently outperforms both single-step and multi-step RAG baselines on every dataset. Importantly, our HGMEM with Qwen2.5-32B-Instruct matches or even outperforms baselines powered by the stronger GPT-4o, underscoring its value in resource-efficient scenarios.

The baselines exhibit mixed performance patterns reflecting their respective representational strengths. For instance, HippoRAG v2 relies on knowledge triples, which provide strong fact representation but limited coverage of events and plots. As a result, it performs well on NoCha but falls behind NaiveRAG on NarrativeQA. In contrast, GraphRAG and LightRAG excel at building global representations but are weaker at capturing fine-grained details, leading them to outperform other baselines on Prelude and NarrativeQA. The two multi-step RAG methods, which mainly employ working memory to iteratively generate subqueries in a chaining fashion, struggle with sensemaking questions, where integrating higher-order relationships is essential.

In comparison, our HGMEM provides strong compositional representations that span from facts to plots, equipping LLM reasoning with high-order correlations and integrated evidence. This enables it to meet the diverse requirements posed by the evaluation tasks.

## 5.2 PERFORMANCE AT DIFFERENT STEPS

During the execution of our multi-step RAG system, the memory progressively evolves and guides the LLM to proceed with retrieval and reasoning. To inspect the effects of memory evolving over multiple interaction steps, we force the LLM to generate responses at every step for a total of six turns, even if it originally decides to terminate the iteration earlier. Figure 3 presents the performances at different steps using Qwen2.5-32B-Instruct on long narrative understanding tasks. Note that $t=0$ represents the initial step when the target query $\hat{q}$ is used for retrieval. We can observe that

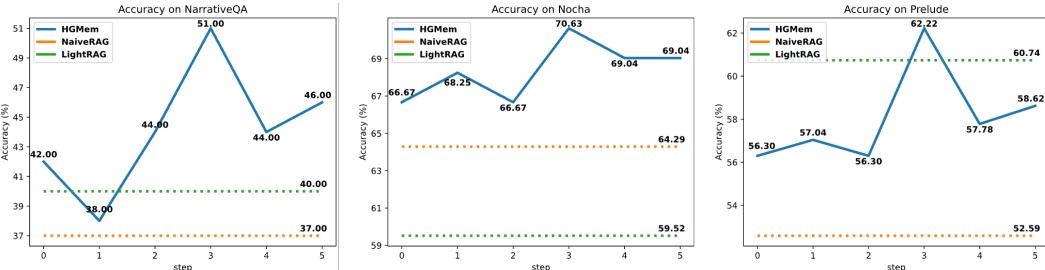

Figure 3: Prediction accuracies at different steps using Qwen2.5-32B-Instruct on long narrative understanding datasets.

Table 2: Ablation results using Qwen2.5-32B-Instruct. "*w/. GE Only*" and "*w/. LI Only*" stand for subquery generation strategies with Global Exploration and Local Investigation, respectively. "*w/o. Update*" and "*w/o. Merging*" refer to HGMEM ablating update and merging operations during memory evolving, respectively.

| Ablation Type | Method | Longbench | | NarrativeQA | Nocha | Prelude |
|---|---|---|---|---|---|---|
| | | Comprehensiveness | Diversity | Acc (%) | Acc (%) | Acc (%) |
| Retrieval Strategy | HGMEM | **64.18** | **66.51** | **51.00** | **70.63** | **62.22** |
| | *w/. GE Only* | 59.25 | 61.67 | 47.00 | 68.25 | 59.26 |
| | *w/. LI Only* | 61.38 | 63.82 | 43.00 | 63.49 | 60.00 |
| Memory Evolution | HGMEM | **64.18** | **66.51** | **51.00** | **70.63** | **62.22** |
| | *w/o. Update* | 62.48 | 64.92 | 50.00 | 68.25 | 60.00 |
| | *w/o. Merging* | 61.76 | 61.80 | 43.00 | 61.11 | 57.78 |

our HGMEM achieves its best performance at $t=3$, mostly outperforming NaiveRAG and LightRAG baselines across steps. More steps bring no further improvements at a higher cost.

## 5.3 ABLATION STUDIES

**Evidence Retrieval Strategy.** When the LLM determines to acquire more information from $\mathcal{D}$ and $\mathcal{G}$, our HGMEM adopts an adaptive memory-based evidence retrieval strategy for either focused local investigation or broad global exploration (Section 3.4). To investigate the effects of such strategy, in Table 2, we compare our strategy to the variants that involve only *Local Investigation* or *Global Exploration*, represented as "*w/. LI Only*" and "*w/. GE Only*", respectively. The results show that both "*w/. LI Only*" and "*w/. GE Only*" significantly underperforms the adaptive strategy across all datasets, demonstrating the effectiveness and necessity of adaptively combining the two modes of evidence retrieval.

**Effects of Update and Merging Operations.** The memory evolving in our HGMEM involves update, insertion and merging operations, where merging is especially critical to building high-order correlations from primitive facts. Because insertion is indispensable, we just carry out ablation experiments on all datasets using Qwen2.5-32B-Instruct to assess the effects of update and merging operations, as shown in Table 2. Compared to the "HGMEM", removing either operation leads to a performance drop, while removing merging ("*w/o. Merging*") causes a substantially larger degradation than removing update ("*w/o. Update*"). It reflects the effectiveness of both operations, especially highlighting the importance of high-order correlations built through merging operations.

## 5.4 DISSECTING QUERY RESOLVING: PRIMITIVE VS. SENSE-MAKING

To better understand how our proposed HGMEM brings improvement to the evaluation tasks, we conduct a targeted analysis across different query types. Specifically, we randomly sample 40 queries from each long narrative understanding dataset used in our experiments, yielding a total of 120 queries. These are then manually categorized into two representative types:

- *Primitive Query*: Queries that primarily require locating directly associated chunks, which can often be resolved with local evidence and focus on straightforward factual information.

Table 3: Average number of entities per hyperedge ($Avg$-$N_v$) in final memory and prediction accuracy (Acc) for a subset of 120 sampled primitive and sense-making queries.

| Query Type | Method | NarrativeQA | | Nocha | | Prelude | |
|---|---|---|---|---|---|---|---|
| | | $Avg$-$N_v$ | Acc (%) | $Avg$-$N_v$ | Acc (%) | $Avg$-$N_v$ | Acc (%) |
| Primitive | HGMEM | 3.35 | 70.00 | 3.78 | 60.00 | 3.85 | 55.00 |
| | *w/o. Merging* | 3.32 | 70.00 | 3.42 | 65.00 | 3.73 | 60.00 |
| Sense-making | HGMEM | 7.07 | 40.00 | 7.97 | 70.00 | 5.25 | 60.00 |
| | *w/o. Merging* | 4.10 | 30.00 | 3.80 | 60.00 | 3.74 | 55.00 |

- *Sense-making Query*: Queries that require deeper comprehension by connecting and integrating multiple pieces of evidence, emphasizing the construction of higher-order relationships and interpretation beyond surface retrieval.

We compare both prediction accuracy and the average number of entities per hyperedge ($Avg$-$N_v$) in memory before generating final responses. The latter serves as a quantitative indicator of relationship complexity. Table 3 shows that on *sense-making queries*, our full "HGMEM" achieves higher accuracy with considerably larger $Avg$-$N_v$ than "HGMEM *w/o. Merging*", demonstrating that forming higher-order correlations enhances comprehension. In contrast, for primitive queries, "HGMEM" yields comparable or slightly lower accuracy relative to "HGMEM *w/o. Merging*". This is likely because the full model still tends to associate additional pieces of relevant evidence (as indicated by the slightly higher $Avg$-$N_v$), even though the primitive evidence alone is sufficient to answer straightforward queries, resulting in redundancy.

Notably, the $Avg$-$N_v$ on sense-making queries consistently exceeds that on primitive queries, especially when merging is applied. Taken together, these results indicate that HGMEM improves contextual understanding by constructing high-order correlations for complex relational reasoning, rather than relying on shallow accumulation of surface facts.

## 6    CONCLUSION

In this work, we propose HGMEM, the hypergraph-based memory mechanism that aims at improving multi-step RAG by enabling the evolving of memory into more sophisticated forms for complex relational modeling. In HGMEM, the memory is structured as a hypergraph composed of a set of hyperedges as separate memory points. HGMEM allows the memory to progressively establish high-order correlations among previously accumulated primitive facts during the execution of multi-step RAG systems, guiding LLMs to organize and connect thoughts for a focal problem. Extensive experiments and in-depth analysis validate the effectiveness of our method over strong RAG baselines on challenging datasets featuring global sense-making questions over long context.

## 7    REPRODUCIBILITY STATEMENT

To ensure reproducibility, we introduce the usage and statistics of our used datasets in Section 4.1 and Appendix A. We also give the implementation details about the offline graph construction, system deployment and configuration in Section 4.2. Appendix D gives the prompts for updating, inserting and merging memory points for memory evolving during multi-step RAG execution. Appendix E describes the procedures for subquery generation with detailed prompts. Appendix F gives the evaluation prompts for scoring model responses in the generative sense-making QA task.

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

Table 4: Statistics of data used in our experiments. #Documents, Avg. #Tokens and #Queries represent the number of documents, average tokens per document and the total number of queries, respectively.

|  | Longbench (Financial) | Longbench (Governmental) | Longbench (Legal) | NarrativeQA | Nocha | Prelude |
|---|---|---|---|---|---|---|
| #Documents | 20 | 22 | 7 | 10 | 4 | 5 |
| Avg. #Tokens | 266k | 256k | 194k | 218k | 139k | 280k |
| #Queries | 100 | 98 | 55 | 100 | 126 | 135 |

## A   DATASET STATISTICS

**Generative Sense-making QA.**   We retain a portion of long documents with more than 100k tokens from **Longbench V2** (Bai et al., 2025), which was originally comprised of six major task categories designed to assess the ability of LLMs to handle long-context problems. In our experiments, we select three domains of documents from the category of single-document QA, including *Financial*, *Governmental* and *Legal*.

**Long Narrative Understanding.**   We use the following public benchmarks:

- **NarrativeQA** (Kociský et al., 2018): It is one of the most widely used benchmarks for story question answering. Because of its question construction strategy over high-level book summaries, the task places greater emphasis on synthesis and inference beyond local texts. In contrast, many other existing long-context QA tasks can often be solved with only local evidence, as shown by studies in (Yu et al., 2025). For evaluation, we randomly sample 10 long books exceeding 100k tokens, together with their associated queries, from the complete benchmark.

- **NoCha** (Karpinska et al., 2024): The task involves discriminating minimally different pairs of true and false claims about English fictional books. Although the format may appear different from sense-making questions, NoCha is explicitly designed to require constructing a global understanding of the book in relation to the focal statement. Since the official test set is hidden, we conduct experiments using only the publicly released subset.

- **Prelude** (Yu et al., 2025): This benchmark assesses LLMs' global comprehension and deep reasoning by requiring them to determine whether a character's prequel story is consistent with the original book. Most instances of this task demand integrating multiple pieces of evidence or even forming a holistic impression of the character's storyline. In our experiments, we use all English books included in Prelude for evaluation.

Table 4 gives the detailed statistics about the data used in our experiments, including the number of documents, average tokens per document and the total number of queries. Generative sense-making QA task involves documents from Longbench V2 benchmark in *Financial*, *Government* and *Legal* domains. Long narrative understanding task uses NarrativeQA, Nocha and Prelude benchmarks.

## B   COMPARISON BASELINES

In our experiments, we compare our methods to traditional RAG and Multi-step RAG methods. Traditional RAG includes:

- **NaiveRAG** just uses the target query to retrieve a set of text chunks from the document for dealing with queries.

- **GraphRAG** (Edge et al., 2024) constructs knowledge graph from plain-text documents and build a hierarchy of communities of closely related entities before using an LLM to make responses.

- **LightRAG** (Guo et al., 2024) also builds a graph structure and employs a dual-level retrieval strategy from both low-level and high-level evidence discovery.

- **HippoRAG v2** (Gutiérrez et al., 2025) creates a knowledge graph and adopts the Personalized PageRank algorithm with dense-sparse integration of passages into the graph search process for resolving queries.

You are an intelligent assistant responsible for resolving the [Main Query] through analyzing supportive information from external knowledge sources and making necessary treatment.
Your current [Memory] records the existing memory points describing what you have already known with respect to the [Main Query].

At present, in order to ultimately resolve the [Main Query], one or several auxiliary subqueries have been raised, i.e. those in [Current Subqueries].
Correspondingly, [Retrieved Info] that contains possibly useful content retrieved by either [Main Query] or [Current Subqueries] in the form of a csv table.

-Goal-
Given the [Retrieved Info], your task is to extract useful information worth memorizing for dealing with the [Main Query] by adding new memory points and/or editing the existing memory points.

-Steps-
1. Based on your existing [Memory], identify useful content worth memorizing from the [Retrieved Info] to better deal with the [Main Query], then reorganize your [Memory] using one or more of the following prescribed operations.
- (1) **Insert New Memory Point(s)**. The Insertion operation should be evoked when some aspects of the identified information are suitable to be inserted into your memory as one or multiple additional points.
- (2) **Update Existing Memory Point(s)**. The Update operation should be evoked when some aspects of the identified information are closely related to existing memory points so that they are suitable to be absorbed into one or multiple existing memory points.

For each inserted or updated memory point, use **{record_delimiter}** as the delimiter to format as
(point{tuple_delimiter}<related_object_1>{object_delimiter}<related_object_2>{object_delimiter}<related_object_3>{tuple_delimiter}<point_description>)

2. Output in {language} as two separate lists of inserted and updated memory points as the **Example of Anticipated Output Format**.
- For memory insertion, just give the newly-inserted memory points in [Inserted Memory Points]. When finished, output {completion_delimiter}.
If there's no meaningful memory points to insert that can bring new information besides current [Memory], just output <None> in [Inserted Memory Points].
- For memory update, indicate the indices of existing memory points to be updated and give the newly-updated memory point(s) in [Updated Memory Points]. When finished, output {completion_delimiter}.
If there's no existing memory points, just output <None> in [Updated Memory Points].
The [Main Query], [Memory], [Current Subqueries] and [Retrieved Info] are given as below. Please output your results as the **Example of Anticipated Output Format**.

######################-Example of Anticipated Output Format-######################
[Inserted Memory Points]:
(point{tuple_delimiter}Alex{object_delimiter}Jordan{object_delimiter}Cruz{tuple_delimiter}Alex and Jordan's shared commitment to discovery highlights their camaraderie and rebellion against Cruz's control, creating a bond based on innovation and mutual goals. Cruz represents an opposing force with a 'narrowing vision' of control, contrasting with the desire for discovery and innovation expressed by Alex and Jordan.){record_delimiter}
(point{tuple_delimiter}Sam Rivera{object_delimiter}Alex{tuple_delimiter}The collaboration between Sam and Alex represents two facets of humanity's response to the unknown intelligence, both driven by their emotional experiences and their acknowledgment of the historical significance of their actions during this first contact situation.){record_delimiter}
{completion_delimiter}

[Updated Memory Points]:
0, (point{tuple_delimiter}Steve Jobs{object_delimiter}San Francisco, California{object_delimiter}Paul{object_delimiter}Clara Jobs{object_delimiter}{tuple_delimiter}Steve Jobs was born on February 24, 1955, in San Francisco, California, and was adopted by Paul and Clara Jobs. He grew up in Mountain View, California, in what would later become Silicon Valley.){record_delimiter}
2, (point{tuple_delimiter}Pricing Strategy{object_delimiter}Premium Pricing{object_delimiter}Freemium Model{object_delimiter}Tiered Pricing Structures{tuple_delimiter}Apple's pricing strategy has evolved from a focus on premium, high-end products with a "cult following" to a more diversified approach that includes both premium and more affordable options like freemium model. This shift has involved strategies like price skimming for new releases, tiered pricing structures with various models at different price points, and even some instances of underpricing to attract new users.){record_delimiter}
{completion_delimiter}

######################-Real Data-######################
[Main Query]: {main_query}

[Memory]:
{memory}

[Current Subqueries]:
{cur_subqueries}

[Retrieved Info]:
{retrieved_info}
######################
Note that:
1. The so-called useful information are those that could enrich the query-specific knowledge or potentially bring insights to better deal with the [Main Query].
2. For each memory point, you should comprehensively organize and summarize its description from the whole context of the [Retrieved Info], rather than just repeat original contents from the given csv table.
3. A memory point can involve multiple closely associated objects so that it can describe higher-level relationships among multiple mutually-connected objects.
4. Meanwhile, avoid forcibly merging everything into a single point. If several distinct objects are not strongly associated, output as separate memory points.
5. Avoid forcibly inserting or updating existing memory points if not necessary.
6. It is encouraged to directly use the existing entity terms listed in the [Retrieved Info] for manipulating memory points. If necessary, you can also introduce new entity terms besides those already explicitly listed.

Output:

Figure 4: The prompt for updating and inserting memory points during memory evolving in HG-MEM.

For resolving the [Main Query], you have consolidated some memory points in your [Memory] recording the relevant information you have known.
Based on current [Memory], your task is to conduct memory reorganization that merges multiple memory points into new ones when they are more suitable to constitute a semantically/logically cohesive unit as a whole.

Specifically, you need to specify the indices of original memory points to merge.
Then, for each newly merged point, provide updated descriptions that could build essentially higher-order associations while preserving their original information necessary for dealing with the [Main Query].

Format each reorganized memory point as <indices>{tuple_delimiter}<new description>
Output in [Points_to_Merge] using {language} as the **Example of Anticipated Output Format**.

######################-Example of Anticipated Output Format-######################
[Points_to_Merge]:
(1,2{tuple_delimiter}<new_description>){record_delimiter}
(2,4,5{tuple_delimiter}<new_description>){record_delimiter}
{completion_delimiter}

######################-Real Data-######################
[Main Query]: {main_query}
[Memory]:
{memory}

######################
Note that, after reorganization,
(1) Each new memory point should encapsulate a semantically/logically cohesive unit that highlights unique and essential association among the involved entities of original memory points.
(2) Each memory point aims to cover distinct aspects, minimizing overlap across different memory points.
(3) Memory redundancy is reduced by eliminating duplicate content across different memory points.
(4) If an original point itself is more suitable to be kept as a separate point, just leave it unchanged and do not output it.
(5) If a memory point has included objects significantly more than other points and encapsulated comprehensive content, it should not be further merged.
(6) Avoid forcibly merging. If there is no suitable points to merge, output <None> in [Points_to_Merge] without any other thing.
(7) Principally, new memory points should primarily reuse original entities and preserve original details as much as possible.

Output:

Figure 5: The prompt for merging memory points during memory evolving in HGMEM.

Table 5: Statistics of the cost of online multi-step RAG execution in our HGMEM and other baselines with working memory. *Avg*-Token is the average count of tokens processed by LLMs per question, while *Avg*-Time stands for the average inference latency per question.

| Method | NarrativeQA | | Nocha | | Prelude | |
|---|---|---|---|---|---|---|
| | *Avg*-Token | *Avg*-Time | *Avg*-Token | *Avg*-Time | *Avg*-Token | *Avg*-Time |
| HGMEM | 4436.43 | 15.84 | 5252.73 | 18.76 | 5421.74 | 19.36 |
| *w/o. Merging* | 4154.02 | 14.84 | 4750.32 | 16.97 | 4897.81 | 17.49 |
| DeepRAG | 3904.18 | 13.94 | 4724.07 | 16.87 | 4514.66 | 16.12 |
| ComoRAG | 5083.26 | 18.15 | 5503.98 | 19.66 | 7827.56 | 27.96 |

Multi-step RAG includes:

- **DeepRAG** (Guan et al., 2025) conducts multi-step reasoning as a Markov Decision Process by iteratively decomposing queries.

- **ComoRAG** (Wang et al., 2025) undergoes multi-step interactions with external data sources with a dynamic memory workspace, iteratively generateing probing queries and integrating the retrieved evidence into a global memory pool.

## C    COST COMPARISON

We conduct a cost comparison between our HGMem and other baselines with working memory in terms of token consumption and inference latency. Note that the cost of online multi-step RAG execution is the real concern for fair comparison because the offline graph construction is just for building query-agnostic indexing structure. With this focus, we measure the average token consumption and inference latency of HGMEM, ComoRAG and DeepRAG in Table 5. From the statistics, we can observe that the cost of our HGMem is basically of the same level with those of DeepRAG and ComoRAG while consistently achieving better performance. We can also see that the merging operation, which is the core operation for forming high-order correlation in our HGMem, just introduces minor computational overhead.

## D    PROMPTS FOR MEMORY EVOLVING

Section 3.5 describes the dynamics of memory evolving in HGMEM, which consists of update, insertion and merging operations. The prompts for these three types of operations are given in Figure 4 and Figure 5.

## E    PROMPTS FOR SUBQUERY GENERATION

During our multi-step RAG execution, the LLM needs to generate subqueries for acquiring information from external data sources. First, it raises relevant concerns that either target at specific memory points or aim at probing useful information outside current memory. Then, the LLM generates corresponding subqueries according to the raised concerns. The prompts for raising concerns and generating subqueries are given in Figure 6 and Figure 7, respectively.

## F    EVALUATION PROMPTS FOR GENERATIVE SENSE-MAKING QA

For the evaluation of generative sense-making QA, we leverage GPT-4o as an evaluator to assess the quality of model responses. Given the target query and the source paragraph from which the query originated, the GPT-4o evaluator first indicates the level of comprehensiveness/diversity and then gives a final score within the value range of the corresponding level. Detailed prompts for such LLM-as-a-Judge evaluation. Figure 8 and Figure 9 give the prompts for scoring the comprehensiveness and diversity, respectively.

You are an intelligent assistant responsible for dealing with the [Main Query] by making appropriate operations as specified.
With respect to the [Main Query], you have consolidated some memory points in your [Memory] describing what you have already known regarding the [Main Query].
Each memory point can be seen as a specific aspect relevant to the [Main Query], providing necessary details or insights from its perspective.

-Goal-
Your task is to analyze the [Main Query] and [Memory], then determine whether current [Memory] has been sufficient to comprehensively resolve the [Main Query].
If not sufficient, you need to indicate what you want to further investigate.

-Procedures-
Step 1.
Make appropriate judgement following the logic branches below.
Case 1: If the [Memory] has been sufficient to completely resolve the [Main Query], output <None> in [Concerns].
Case 2: If the [Memory] is not sufficient, determine current situation should be attributed to which of the following subcases.
    Case 2.1: There are some specific memory points which you want to further investigate more details about.
    Case 2.2: There are unexplored aspects that go beyond the scope of current [Memory] (i.e. not related to any of the existing memory points).

Step 2.
Output as **Example of Anticipated Output Format**.
Specifically, give your judgement in [Judgement] using corresponding case index (1, 2.1 or 2.2).
Then, generate several concerns that aim at exploring details or aspects not addressed by current [Memory] to better resolve the [Main Query]
    When case 2.1, generate up to {num_concerns} concerns, each of which targets at a specific memory point. For each concern, specify the index of its corresponding memory point.
    When case 2.2, generate up to {num_concerns} concerns that probe meaningful information not yet covered by current [Memory]
##########-Example of Anticipated Output Format for Case 1-##########
[Judgement]: 1
[Concerns]: <None>

##########-Example of Anticipated Output Format for Case 2.1-##########
[Judgement]: 2.1
[Concerns]:
0{tuple_delimiter}your_concern_1{record_delimiter}
2{tuple_delimiter}your_concern_2{record_delimiter}
2{tuple_delimiter}your_concern_3{record_delimiter}
{completion_delimiter}

##########-Example of Anticipated Output Format for Case 2.2-##########
[Judgement]: 2.2
[Concerns]:
your_concern_1{record_delimiter}
your_concern_2{record_delimiter}
your_concern_3{record_delimiter}
{completion_delimiter}

#####################-Real Data-#####################
[Main Query]: {query}

[Memory]:
{memory}
#####################
* Note that:
(1) Your concern should be concise and suggest what further details or aspect you subsequently will seek for.
(2) Only output the judgement, concerns, and the indices of corresponding memory points without any other content.
(3) If current [Memory] has covered most relevant perspectives, generate fewer concerns to avoid redundancy.
(4) Your generated concerns should be separated by "{record_delimiter}".

#####################
Output:

Figure 6: The prompt for raising concerns either targeting at specific memory points or probing useful information outside the current memory.

You are an assistant responsible for dealing with the [Main Query].
Although you have had some relevant information in your [Memory], your current [Memory] is still not sufficient to comprehensively resolve the [Main Query] due to the concern given in [Concern].
Therefore, you need to generate a subquery that aims at either retrieving more evidences or investigating unexplored aspects in [Subquery] to better deal with the [Main Query] ultimately.

[Previous Subqueries] records a series of previous subqueries that have been raised before.

##########-Anticipated Output Format-##########
[Subquery]: xxx

#####################-Real Data-#####################
[Main Query]: {query}

[Memory]:
{memory}

[Concern]:
{concern}

[Previous Subqueries]:
{history_subqueries}

#####################
* Note that:
(1) Your generated subquery should be concise and address the concerns in your [Concern].
(2) You should avoid generating a subquery that is overly similar to any one of the [Previous Subqueries] or [Main Query].
(3) Only output your subquery without any other redundant content such as markup strings.
#####################
Output:

Figure 7: The prompt for generating subqueries based on previously raised concerns.

Given a [Paragraph] and a [Question], you will evaluate the quality of the [Response] in terms of Comprehensiveness.

######################-Real Case-######################
[Paragraph]:{paragraph}
[Question]: {question}
[Response]:{response}

######################-Evaluation Criteria-######################
Comprehensiveness measures whether the [Response] comprehensively covers all key aspects in the [Paragraph] with respect to the [Question].
Level  | score range | description
Level 1 | 0-20  | The response is extremely one-sided, leaving out key parts or important aspects of the question.
Level 2 | 20-40 | The response has some content, but it misses many important aspects of the question and is not comprehensive enough.
Level 3 | 40-60 | The response is moderately comprehensive, covering the main aspects of the question, but there are still some omissions.
Level 4 | 60-80 | The response is comprehensive, covering most aspects of the question, with few omissions.
Level 5 | 80-100 | The response is extremely comprehensive, covering almost all aspects of the question no omissions, enabling the reader to gain a complete and thorough understanding.
Evaluate the [Response] using the criteria listed above, give a level of comprehensiveness in [Level] based on the description of the indicator, then give a score in [Score] based on the corresponding value range, and finally explain in [Explanation].

Note that:
(1) You should reference to the [Paragraph] and avoid misinterpreting any content of [Paragraph] as part of the [Response].
(2) Avoid excessively concerning very specific details. When the response mentions an aspect without providing very specific details, you should consider this aspect as validly covered, as long as the omitted detail is not crucial to particularly mention with respect to the [Question] in the whole scope of the response.
(3) If [Response] contains extra content not directly included in the [Paragraph], as long as the extra content is correct, do not consider the extra content as defects for giving final evaluation.
(4) You should conform to the -Anticipated Output Format- and give your evaluation results in [Your Evaluation].
######################-Anticipated Output Format-######################
[Level]: A level ranging from 1 to 5  # This should be a single number, not a range.
[Score]: A value ranging from 0 to 100  # This should be a single number satisfying the ranging constraint of the corresponding [Level], not a range.
[Explanation]: xxx
[Your Evaluation]:

Figure 8: The prompt for evaluating the comprehensiveness of a model response.

Given a [Paragraph] and a [Question], you will evaluate the quality of the [Response] in terms of Diversity.

######################-Real Case-######################
[Paragraph]: {paragraph}
[Question]: {question}
[Response]: {response}

######################-Evaluation Criteria-######################
Diversity measures how varied and rich is the response in offering different perspectives and insights related to the question.
Level  | score range | description
Level 1 | 0-20  | The response is extremely narrow and repetitive, providing only a single perspective or insight without exploring alternative viewpoints or additional information.
Level 2 | 20-40 | The response offers a few different perspectives but remains largely superficial. It may touch on alternative viewpoints but does not elaborate or provide substantial insights.
Level 3 | 40-60 | The response moderately presents several perspectives with moderate depth. It begins to integrate different viewpoints and insights but may still miss some important angles or lack thorough exploration.
Level 4 | 60-80 | The response is rich in perspectives and insights. It basically explores multiple viewpoints and provides substantial evidence and examples to support each angle.
Level 5 | 80-100 | The response is exceptionally varied and rich in perspectives and insights. It offers a comprehensive exploration of the question, addressing multiple angles with depth and originality.
Evaluate the [Response] using the criteria listed above, give a level of comprehensiveness in [Level] based on the description of the indicator, then give a score in [Score] based on the corresponding value range, and finally explain in [Explanation].

Note that:
(1) You should reference to the [Paragraph] and avoid misinterpreting any content of [Paragraph] as part of the [Response].
(2) If [Response] contains extra content not directly included in the [Paragraph], as long as the extra content is correct, do not consider the extra content as defects for giving final evaluation.
(3) You should conform to the -Anticipated Output Format- and give your evaluation results in [Your Evaluation]
######################-Anticipated Output Format-######################
[Level]: A level ranging from 1 to 5  # This should be a single number, not a range.
[Score]: A value ranging from 0 to 100  # This should be a single number satisfying the ranging constraint of the corresponding [Level], not a range.
[Explanation]: xxx
[Your Evaluation]:

Figure 9: The prompt for evaluating the diversity of a model response.

# G CASE STUDY

As shown in Table 6, we present two representative cases highlighting HGMem's distinct reasoning advantages over LightRAG from the perspective of forming high-order correlations and the strategy of adaptive memory-based evidence retrieval during memory evolving.

The first case is from NarrativeQA, where the question requires inferring the underlying cause of Xodar's enslavement—a relation not explicitly stated in the original text. LightRAG just makes incorrect surface-level predictions based on the retrieved content. While DeepRAG stores the knowledge in the memory, it does not form high-order correlation and fails to predict correctly. In contrast,

HGMem progressively evolves its memory and establishes high-order correlations from primitive evidences accumulated from past interactions, uncovering that Xodar's punishment originates from his defeat by Carter.

The second case is from Nocha, where the query mixes factual and misleading details. The LLM raises a subquery about the source of the name 'White Sands'. Using the strategy of local investigation, it particularly conducts in-depth inspection about the related memory point (Point 1) in current memory and verifies that there is no clear evidence showing the name was given by Anne. However, LightRAG mistakenly recognizes that the name 'White Sands' was given by Anne and DeepRAG doesn't qualify the correctness of 'White Sands'.

Together, these examples show that HGMem enables a deeper and more accurate contextual understanding beyond superficial text retrieval.

## H  A TOY EXAMPLE

To illustrate the core workflow of our method, we present a toy example in Figure 10. Given the query "Why is Xodar given to Carter as a slave?", the LLM first retrieves relevant evidence, converting it into a structured representation (corresponding to Point 0 in the figure). It then generates sub-queries based on current memory to retrieve missing reasoning elements. In the subsequent iteration, newly retrieved evidence is integrated into the memory storage through update, insertion and merging operations, yielding a unified representation that includes high-order memory points capturing complex relationships beyond surface content in original data sources. Finally, the LLM leverages its evolved memory to produce an answer to the target query. This example illustrates how the memory evolves during the multi-step RAG execution to iteratively refine its understanding and support complex relational modeling.

Table 6: Illustrative Cases on NarrativeQA and Nocha, where red and blue stand for the relevant answer and its corresponding source, respectively

| Source | NarrativeQA | Nocha |
|---|---|---|
| **Question** | Why is Xodar given to Carter as a slave? | Answer TRUE if the statement is true, otherwise answer FALSE
[Statement]: Anne assigns romantic names to her surroundings, like "Lake of Shining Waters" and "White Sands".
[Question]: Based on the context provided, is the above statement TRUE or FALSE? |
| **Golden Answer** | As his punishment for being defeated by Carter previously | False |
| **Relevent Chunks** | **1.** Xodar was brought from the adjoining room. Issus glared at him, a baleful light in her hideous eyes. "And such as you are a Dator of the First Born?" she squealed. "For the disgrace you have brought upon the Immortal Race, you shall be degraded to a rank below the lowest. No longer be you a Dator, but forevermore a slave of slaves...."
**2.** ... | **1.** "Oh, I don't like that name, either. I shall call it—let me see—the Lake of Shining Waters. Yes, that is the right name for it. I know because of the thrill. When I hit on a name that suits exactly, it gives me a thrill. Do things ever give you a thrill?" Matthew ruminated. ..."
**2.** said Anne dreamily. "Is it as nice as it sounds? Just when you said 'shore road', I saw it in a picture in my mind, as quick as that! And White Sands is a pretty name, too, but I don't like it as well as Avonlea. Avonlea is a lovely name. It just sounds like music. ... |
| **LightRAG Retrieval&Prediction** | **Prediction:**
Xodar is given to Carter as a slave due to his degradation by Issus after falling from favor. Issus degrades Xodar to a rank below the lowest and commands that he serve as a slave to Carter, who previously conquered Xodar in the prison on the Isle of Shador. | **Prediction:**
[Answer]: TRUE
[Explanation]: The statement is true based on the information provided in the data tables. In the Relationships table, there is a record that indicates Anne of Green Gables gave the romantic name "Lake of Shining Waters" to Barry's Pond. Additionally, in the Sources table, Anne expresses a preference for the name "Avonlea" over "White Sands", suggesting she has a fondness for romantic names. |
| **DeepRAG Retrieval&Prediction** | **Generated Subquery:**
**1.** What factors influenced Issus's decision to demote Xodar from a Dator to a slave?
**2.:** What specific orders or instructions did Issus give to ensure Xodar was transferred to Carter as a slave?
– – – – – – – – – – – – – – – – – – – – – – – – – – – – – – –
**Memory:**
**1.** Xodar, originally a member of the First Born, was degraded to a slave status by Issus due to his fall from favor. This demotion signifies...
**2.** Issus, in a display of her absolute authority, degrades Xodar to the lowest status of a slave, specifically instructing him to serve Carter. This act is designed to further demean Xodar and to test Carter's response.
– – – – – – – – – – – – – – – – – – – – – – – – – – – – – – –
**Prediction:**
Xodar is given to Carter as a slave because Issus seeks to degrade and humiliate him. This act undermines Xodar's former prestige and weakens his influence, while tying Carter's treatment and success to the actions of a once-respected figure, likely intended to provoke internal conflict and dissatisfaction. | **Generated Subqueries:**
**1.** Is there evidence in the provided memory or data that Anne has given romantic names to other locations besides Barry's Pond?
**2.** Does the extended info suggest that Anne applies her romantic naming to all her surroundings?
– – – – – – – – – – – – – – – – – – – – – – – – – – – – – – –
**Memory:**
**1.** Anne gives the romantic name "Lake of Shining Waters" to Barry's Pond, indicating her imaginative and poetic nature towards her surroundings.
**2.** Anne finds Lover's Lane picturesque and romantic, appreciating its natural beauty and atmosphere for thoughtful reflection. This highlights her imaginative and poetic nature towards her surroundings.
– – – – – – – – – – – – – – – – – – – – – – – – – – – – – – –
**Prediction:**
[Answer]: TRUE
[Explanation]: The statement is true. There is a record that indicates Anne of Green Gables gave the romantic name "Lake of Shining Waters" to Barry's Pond, indicating her imaginative and poetic nature towards her surroundings. |
| **HGMEM Retrieval&Prediction** | **Generated Subquery:**
**1.** What specific incident of Xodar was considered a disgrace to the Immortal Race, leading directly to his punishment by Issus? *(Global Exploration)*
**2.:** What role does Carter have in the decision to give Xodar to him as a slave? *(Global Exploration)*
**3.**....
– – – – – – – – – – – – – – – – – – – – – – – – – – – – – – –
**Original Memory:**
**Point I:**
*Subordinate Entities*: Xodar, Issus, Cowardly injustice
*Description*: Xodar is punished by Issus as a form of cowardly injustice for the disgrace he has brought upon the Immortal Race.
**Point II:**
*Subordinate Entities*: Xodar, Issus, Degradation
*Description*: Issus degrades Xodar to a rank below the lowest, Xodar is no longer a Dator, but forever a slave of slaves, to fetch and carry for the lower orders that serve in the gardens of Issus.
– – – – – – – – – – – – – – – – – – – – – – – – – – – – – – –
**Memory After Evolving** (Guided by Subquery 1):
**Point I** (*Merged*):
*Subordinate Entities*: Xodar, Issus, Cowardly injustice, Degradation
*Description*: Xodar's degradation into a slave is a result of his disgraceful defeat by John Carter, which brought shame to the Immortal Race, leading Issus to punish him as a form of cowardly injustice. This reflects the societal judgment and loss of honor within the Immortal Race's culture.
– – – – – – – – – – – – – – – – – – – – – – – – – – – – – – –
**Prediction:**
Xodar is given to Carter as a slave due to his disgraceful defeat by John Carter, which brought shame to the Immortal Race. Issus, the goddess, punishes Xodar for his cowardice, degrading him to a slave, and he is subsequently treated as such by Carter. | **Generated Subqueries:**
**1.:** Is there any evidence in "Anne of Green Gables" that suggests Anne gave a romantic name to a place known as White Sands? *(Local Investigation)*
**2.:** What's the actual name of White Sands in the book "Anne of Green Gables"? *(Local Investigation)*
**3.**....
– – – – – – – – – – – – – – – – – – – – – – – – – – – – – – –
**Original Memory:**
**Point I:**
*Subordinate Entities*: Diana, Barry's pond, Lover's lane, Gilbert, Anne, White sands, Anne of Green Gables
*Description*: ... Anne has a habit of giving romantic names to her surroundings, such as calling Barry's Pond the "Lake of Shining Waters". While she does romanticize places, there is no specific mention of her assigning a romantic name to "White Sands".
– – – – – – – – – – – – – – – – – – – – – – – – – – – – – – –
**Memory after Evolving** (Guided by Subquery 1):
**Point I** (*Updated*):
*Subordinate Entities*: Diana, Barry's pond, Lover's lane, Gilbert, Anne, White sands, Anne of Green Gables
*Description*: ...Anne has a habit of giving romantic names to her surroundings, such as calling Barry's Pond the "Lake of Shining Waters". However, there is no specific mention of her assigning a romantic name to "White Sands". White Sands is instead associated with Gilbert and an educational opportunity offered to Anne by the White Sands trustees.
– – – – – – – – – – – – – – – – – – – – – – – – – – – – – – –
**Prediction:**
[Answer]: FALSE
[Explanation]: According to the provided information, Anne does give a romantic name to Barry's Pond, referring to it as the "Lake of Shining Waters". However, there is no evidence in the memory or data tables that Anne assigns a romantic name to "White Sands". Instead, White Sands is associated with Gilbert and an educational opportunity for Anne, and it is mentioned as a geographical location without any indication that Anne romanticizes it. Therefore, the statement is false because it incorrectly includes "White Sands" as one of the locations Anne romanticizes. |

**Query**: Why is Xodar given to Carter as a slave?

---

**[Iteration 0]**

**Memory** $M_0$: None

↓ Retrieve Information

Memory Evolving

Generate Subquery

---

**[Iteration 1]**

**Memory** $M_1$:
- Point I

*Subordinate Entities:* Xodar, Issus, Cowardly Injustice
*Description:* Xodar is punished by Issus as a form of cowardly injustice for the disgrace he has brought upon the Immortal Race.

- Point II

*Subordinate Entities:* Xodar, Issus, Degradation
*Description:* Issus degrades Xodar to a rank below the lowest, Xodar is no longer a Dator, but forever a slave of slaves, to fetch and carry for the lower orders that serve in the gardens of Issus.

- Point III

*Subordinate Entities:* Nobel Blacks, Xodar, Thurid
*Description:* The Noble Blacks exhibit hostility towards Xodar, reviling and cursing him for his fall from favor. Thurid is suggested as a challenge for Xodar, demonstrating the social dynamics and conflicts within the group. This reflects the broader theme of societal judgment and persecution against those who have lost status.

↓ Retrieve Information

Memory Evolving

---

**[Iteration 2]**

**Memory** $M_2$:
- Point I

*Subordinate Entities:* Xodar, Issus, Cowardly Injustice,Degradation
*Description:* Xodar's degradation into a slave is a result of his disgraceful defeat by John Carter, which brought shame to the Immortal Race}, leading Issus to punish him as a form of cowardly injustice. This reflects the societal judgment and loss of honor within the Immortal Race's culture.

- Point II

*Subordinate Entities:* Nobel Blacks, Xodar, Thurid
*Description:* The Noble Blacks exhibit hostility towards Xodar, reviling and cursing him for his fall from favor. Thurid is suggested as a challenge for Xodar, demonstrating the social dynamics and conflicts within the group. This reflects the broader theme of societal judgment and persecution against those who have lost status.

↓ Generate Response

---

**Response:** Xodar is given to John Carter as a slave by Issus to degrade him further after his defeat and humiliation. Issus wants to punish Xodar by making him serve the very person who conquered him, marking a drastic change in Xodar's status from a member of the First Born to a slave.

**Reference Answer:** As his punishment for being defeated by Carter previously

Figure 10: A toy example of HGMem workflow from the NarrativeQA dataset of GPT-4o

