# OpenReview forum: "Improving Multi-step RAG with Hypergraph-based Memory"
_ICLR.cc/2026/Conference — Submitted to ICLR 2026_

### Official Review · Reviewer_Xaut · 2025-10-28

**Soundness:** 2
**Presentation:** 2
**Contribution:** 2
**Rating:** 4
**Confidence:** 3

**Summary:**

This paper proposes HGMem, a hypergraph-based memory for multi-step RAG that captures high-order relationships among facts, enabling dynamic, structured knowledge evolution. By moving beyond passive storage, HGMem enhances global understanding and reasoning, outperforming strong baselines on challenging sense-making tasks.

**Strengths:**

* Long context understanding is an interesting topic
* The paper is well written

**Weaknesses:**

* The authors claim to address multi-step RAG; however, the focus of the work appears to lean more toward long-context understanding. Notably, the experimental evaluation does not include standard multi-hop RAG benchmarks such as HotpotQA. While I am not necessarily advocating for additional experiments on more datasets, the paper may benefit from a clearer framing or adjustment of its stated objectives to better align with the actual contributions.
* The paper seems to be more concerned about long context understanding, and try to use hypergraph to build memory about the context, but this field has been well studied [1,2,3].
* I am kindly worried about the latency as the method requires to update the hypergraph during reasoning and the update requries additional call of the LLM


[1]Xu W, Mei K, Gao H, et al. A-mem: Agentic memory for llm agents[J]. arXiv preprint arXiv:2502.12110, 2025.

[2]Ong K T, Kim N, Gwak M, et al. Towards lifelong dialogue agents via timeline-based memory management[J]. arXiv preprint arXiv:2406.10996, 2024.

[3]Rasmussen P, Paliychuk P, Beauvais T, et al. Zep: a temporal knowledge graph architecture for agent memory[J]. arXiv preprint arXiv:2501.13956, 2025.

**Questions:**

* I am actually a little confused about what will the hypergraph be like given a query and correspondding context, how could it help the model to better understand the context
* How will the hypergraph be inputted into the LLM? Can the model effectively understand the graph, the experiments are conducted on relatively strong models, what about on those weaker models like Qwen 7B or Llama 8B?
* In section 4.2, the authors build a offline graph, what is it used to do? Simply for baseline like lightRAG or it is also used for HGMEM?

---

> ### Author Response · Authors · 2025-11-26
> **Response to Reviewer Xaut (Part One)**
>
> # Response to your Weakness 1: Research Focus, Evaluation Setting, Adjust Paper Framing
>
> ***As suggested by your kind concern, we have revised the introduction section to provide a clearer framing about the stated objectives of our paper.***
>
> The primary objective of our work is to address the limitation and challenge of current multi-step RAG methods under long-context understanding scenarios, especially global sense-making tasks. This kind of tasks often require the model to reason over long contexts (even the entire documents) to achieve accurate and deep comprehension, rather than just precise fact retrieval. We have reframed such objectives and task scopes in the updated Introduction section (See Lines 56-61).
> Under such setting, HotpotQA is not an appropriate dataset for validation experiments due to the following two aspects:
>
> (1) HotpotQA is intrinsically a fact-oriented multi-hop reasoning task that just needs locating factual evidences across documents, whose questions can be typically decomposed into clear subqueries featuring shallow entity retrieval. In contrast, our target tasks should involve sense-making questions that necessitate deep understanding, where getting correct answers depend on comprehending global contexts (e.g. the whole narrative), rather than merely locally retrieved explicit facts.
>
> (2) HotpotQA was created before the rise of LLMs. Its data are likely to have been included in the pre-training corpora of many advanced LLMs. This raises potential data leakage risk, which makes it less suitable for evaluating recent LLM-based methods that claim memory capabilities. Moreover, those prior RAG-related papers that used HotpotQA rarely focus on the same setting as ours, most of which are still constrained to handle multi-hop reasoning tasks of shallow fact-oriented problems.

---

> ### Author Response · Authors · 2025-11-26
> **Response to Reviewer Xaut (Part Two)**
>
> (Cont'd)
>
> # Response to your Weakness 2: Clarification about "The field of using hypergraph for long context understanding has been well studied"
>
> Even though there have been several previous work adopting graph-like structure to enhance LLMs’ capabilities for long context understanding, it cannot be said that this field has been well studied. There still remain many aspects worth exploring within the task scope of this paper.
>
> First, as discussed in Lines 54-63, most prior methods treat memory narrowly as an external storage, where information is repeatedly inserted or deleted without being structurally integrated or organized into high-order correlation during multi-step RAG.
>
> Besides, for your mentioned work, although they partially address the above aspect by using graph-like structure to organise memory, their usage of memory is significantly different from our proposed HGMem in terms of (1) applied scenario and methodological focus and (2) memory paradigm.
>
> _(1) Different Applied Scenarios and Methodological Focus_
>
> Your mentioned prior studies totally focus on conversational tasks with clear temporal order. Targeting at conversational tasks with clear timelines, the methodological focus of your mentioned studies mostly lies in exploiting the temporal information within conversation sessions. For instance, THEANINE manages memories by linking them to augment RAG with memory timelines based on explicit temporal and cause-effect relation. The memory in Zep is powered by a temporally-aware dynamic knowledge graph.
> However, such prerequisite task characteristic of clear timelines is not prevalent in most existing tasks either, which makes their methods unsuitable to be immediately generalised to other scenarios other than temporal reasoning. Further experiments indeed show that A-Mem performs poorly on our tasks with no timelines. You can refer to the results given at the bottom.
>
> _(2) Different Memory Paradigm_
>
> ***The memory in our HGMem is a short-term working memory rather than long-term persistent memory. You can refer to the second clarification point in our general response to all reviewers for clearer discrimination.***
>
> To empirically demonstrate the advantage of our HGMem, We further compare our HGMem with A-Mem (the most competitive among your mentioned work) on our intended task scenarios.
>
> | Method      | NarrativeQA-Narrative (ACC) | Nocha-Book (ACC) | Prelude-English (ACC) |
> |-----------|------------------------------|-------------------|-------------------------|
> | A-Mem | 47.00        | 65.08                     | 55.56             |
> | HGMem     | 51.00         | 70.63                        | 62.22             |
>
> As shown, A-Mem performs substantially worse than HGMem on NoCha and Prelude—datasets that require complex relational reasoning over long contexts without explicit temporal timelines. This is as expected since A-Mem originally targets at temporally segmented dialogue tasks where answers depend on short local spans and only a small portion of the input needs to be accessed. Such assumptions do not hold in global sense-making tasks within long contexts, as discussed in the preceding point. Summary-based indexing and memory retrieval, as used by A-Mem, are therefore insufficient for capturing these global dependencies. In contrast, HGMem’s dynamically evolving working memory enables the construction of high-order correlations across multiple reasoning steps, leading to the significantly stronger performance observed above.
>
> Overall, the above discussed differences and experimental results reveal that the field of building memory for long-context understanding still has large rooms for improvement, far from being well-studied.

---

> ### Author Response · Authors · 2025-11-26
> **Response to Reviewer Xaut (Part Three)**
>
> (Cont’d)
>
> # Response to your Weakness 3: Latency of our method.
>
> It seems that it might be a misunderstanding about the actual interaction between the LLM and the hypergraph.
>
> As our response to your Questions 1 and 2, when the LLM reads the hypergraph-based memory, the hypergraph is input into the LLM in a structured text form. During the memory evolving, the LLM is just responsible for generating the textual description of the higher-order memory point (hyperedge). The actual topological update on the hypergraph is well supported by the hyperdb package without any LLM calling, which just minorly increase latency. The LLM itself does not directly manipulate the hypergraph update.
>
> ***For detailed cost analysis of our proposed method, please refer to the third clarification point in our general response to all reviewers.***
>
> # Response to Question 1: What will the hypergraph be like, how could it help the model to better understand the context?
>
> ***To provide a clearer visualization of what the constructed hypergraph-based memory is like and how it outperforms other baseline methods, we also added a case study section in Appendix G of the revised version for better illustration.***
>
> Given a query and corresponding context, after multi-step RAG execution, the hypergraph will finally contain a set of hyperedges, each of which serves as an individual memory point. In our revised paper version, as shown in the cases from Figure 3 (Section 3.5) and the toy example in Figure 10 (Appendix H), every memory point contains its subordinate entities and point description presented in the form of pure text. Every subordinate entity corresponds to a vertex while every memory point corresponds to a hyperedge of the memory hypergraph. The whole memory is conceptually equivalent to a hypergraph.
>
> Note that the final hypergraph with two hyperedges (memory points) in Figure 3 are just for illustrating what the hypergraph would be like. There might be more than two hyperedges in the final hypergraph-based memory in most real-world cases.
>
> The hypergraph helps models to better understand contexts from the following aspects:
>
> (1) The hypergraph structure enables the memory to form high-order correlation. Apart from those hyperedges formed by update and insertion operations, the final hypergraph contains some of hyperedges formed by merging several closely related memory points that are more suitable to constitute a single semantically/logically cohesive unit. In this way, the model benefits from the strong propositions provided by such hypergraph structure for deep reasoning, facilitating the resolution of queries that require complicated sense-making with disparate facts not co-located in contiguous text pieces.
>
> (2) At each interaction step,  based on the topology of hypergraph structure, HGMem memory guides the LLM to adaptively generate subqueries for local investigation or global exploration (Section 3.4), connecting facts and thoughts around the focal problem.

---

> ### Author Response · Authors · 2025-11-26
> **Response to Reviewer Xaut (Part Four)**
>
> (Cont’d)
>
> # Response to Question 2: How will the hypergraph be inputted into the LLM? Performance with weaker models?
>
> ***As stated in the response to your Question 1, the whole memory is conceptually equivalent to a hypergraph presented in a structured text form, which can be directly inputted into the context window of LLMs. Such plaintext representation of graph/hypergraph has been applied in various existing LLM applications [1,2,3].***
>
> To further verify the reviewer’s concern about model capacity, we also conduct experiments with smaller models—LLaMA-8B and Qwen-7B—on the NoCha, NarrativeQA, and Prelude datasets.
>
> | Model                     | Method        | Working Memory | NarrativeQA-Narrative (ACC) | Nocha-Book (ACC) | Prelude-English (ACC) |
> |----------------------------|---------------|----------------|-----------------------------|-----------------|----------------------|
> | **Llama-3.1-8B**|               |                |                             |                 |                      |
> |                            | NaiveRAG      | ✗              | 48.00                          | 53.97           | 51.85                |
> |                            | GraphRAG      | ✗              | 49.00                          | 56.35           | 51.11                |
> |                            | LightRAG      | ✗              | 46.00                          | 58.73           | 54.81                |
> |                            | HippoRAG V2   | ✗              | 38.00                          | 56.35           | 46.67                |
> |                            | ComoRAG       | ✗              | 44.00                          | 54.76           | 42.22                |
> |                            | DeepRAG       | ✔              | 49.00                          | 59.52           | 51.85                |
> |                            | **OurRAG**    | ✔              | **51.00**                      | **59.52**       | **56.30**            |
> | **Qwen2.5-8B**  |               |                |                             |                 |                      |
> |                            | NaiveRAG      | ✗              | 39.00                          | 57.14           | 51.85                |
> |                            | GraphRAG      | ✗              | 40.00                          | 58.73           | 51.85                |
> |                            | LightRAG      | ✗              | 39.00                          | 55.56           | 52.59                |
> |                            | HippoRAG V2   | ✗              | 35.00                          | 58.73           | 51.11                |
> |                            | ComoRAG       | ✗              | 34.00                          | 54.76           | 52.59                |
> |                            | DeepRAG       | ✔              | 40.00                          | 57.94           | 48.89                |
> |                            | **OurRAG**    | ✔              | **42.00**                      | **60.32**       | **53.33**            |
>
>
> These results reveal that these smaller LLMs can still benefit from the structured hypergraph representation.
>
> # Response to Question 3: What is the offline graph used to do?
>
> The offline graph is used for both the baseline (LightRAG) and our HGMem.
> This offline graph, built from the original text document, serves as the foundational knowledge source from which all methods retrieve information. For HGMem, this graph is crucial because the online formation of hyperedges relies on retrieving and linking the existing nodes and edges from this pre-built graph.
>
> # References:
>
> [1] Fatemi et al. Talk like a graph: Encoding graphs for large language models. arXiv:2310.04560, 2023.
>
> [2] Ye et al. Language is all a graph needs. Findings of EACL 2024.
>
> [3] Xu et al. A-mem: Agentic memory for llm agents. arXiv:2502.12110.

---

### Official Review · Reviewer_h1Xj · 2025-10-29

**Soundness:** 3
**Presentation:** 2
**Contribution:** 2
**Rating:** 4
**Confidence:** 4

**Summary:**

This paper proposes HGMEM, a hypergraph-based memory mechanism focusing on dynamic scenarios. It helps improve performance under iterative RAG scenarios.

**Strengths:**

The overall motivation of the paper is great. The paper conducts comprehensive experiments on multiple datasets. The experimental details are extensive, and the paper experimentally explores a wide range of aspects of the proposed framework.

**Weaknesses:**

1. I think the description of hypergraph in this paper is not unclear. As the core innovation that distinguishes it from other graph-RAG methods, it should be clearly explained. However, I find it difficult to understand its specific meaning. I hope the authors can provide a clear and symbolic definition in Sections 3.3 and 3.4, rather than redundant statements. It would be helpful to provide more detailed examples and a clearer comparison with the current graph-RAG process.
2. This method does not seem to have a training process. How can a native LLM or retrieval model be adapted to the proposed architecture?
3. Although there are some reproduction instructions, the paper does not seem to provide code.

**Questions:**

See weaknesses above. Please respond to these cons.

---

> ### Author Response · Authors · 2025-11-26
> **Response to Reviewer h1Xj (Part One)**
>
> # Response to Weakness 1: Description of hypergraph is not clear. Provide symbolic definitions in Sections 3.3 and 3.4. Provide more detailed examples and a clearer comparison
>
> To address your concern, we take the following measures to increase the clarity of our descriptions about hypergraph in the revised version of our paper.
> We refine Section 3.3 by adding the symbolic definition of our hypergraph-based memory storage.
> In Section 3.4, we provide formal symbolic description of the adaptive memory-based evidence retrieval strategy, namely local investigation and global exploration.
> We also add a toy example in Appendix H and a case study in Appendix G to make a clearer comparison with current representative multi-step RAG and graph-RAG approaches.
>
> ***Here, to help you understand our work better, we further elaborately explain the difference of our HGMem with current typical graph-RAG approaches.***
>
> _Before elaboration, we kindly hope the reviewers can first read our general response to all reviewers to gain a clearer discrimination between short-term working memory and long-term persistent memory._
> Current graph-RAG frameworks (e.g., GraphRAG, LightRAG) typically rely on an offline-constructed static graph G=(V,E), where edges represent fixed semantic relations between nodes. This static graph serves as a form of long-term memory to support subsequent RAG executions. However, due to the static nature of  their pre-constructed graph structures, their methods mainly focus on efficient traversal and retrieval over fixed indexing structures.
> In contrast, our HGMem introduces an online-evolving hypergraph-based working memory. Thus, instead of relying on a fixed topology, HGMem dynamically updates, inserts and merges hyperedges as reasoning proceeds, allowing the model to form new correlations unseen in the original document. Besides, the hypergraph-based memory also guides the execution of multi-step RAG through adaptive memory-based evidence-retrieval based on its hypergraph topology.
>
> # Response to Weakness 2:  Training-free. How can LLMs be adapted to HGMem?
>
> Yes, our HGMem is not dependent on any training process. The only requirements of our framework is to employ an LLM that can follow prompt instructions specified by the users. As for retrieval model, users can choose any open-source pre-trained embedding models such as BGE-m3, Qwen3 or GPT-emb as long as it can satisfy users’ own requirements for semantic matching. This design is intentional: it allows HGMem to maintain flexibility across different LLMs and retrieval engines, avoiding the need for task-specific retraining. Otherwise, the users would have to collect training data and conduct task-specific fine-tuning, which probably reduces the generalisability of models across different tasks. It is also worth mentioning that such totally training-free methods have been widely adapted to various complex architecture in both academia and industry [1,2,3,4].
>
> ***Here, we further elaborate how we ensure an LLM to effectively follow the designed reasoning and retrieval process in a totally prompt-driven way.***
>
> The prompts for the key operations in our RAG workflow includes:
>
> (1) *Prompts for Subquery Generation* – it prompts the LLM to decompose complex questions into structured subqueries based on the current memory state.
>
> (2) *Prompts for Memory Evolution* – it explicitly defines how retrieved information should be integrated into hypergraph-based memory through update, insertion and merging operations.
>
> Each of these prompt-driven stages employs predefined templates with explicit field delimiters (e.g., [Subquery], [Memory]), ensuring that the model’s outputs can be easily parsed and reliably integrated into HGMem execution. In the revised version of our paper, we have added the prompts used for these necessary operations of our architecture for reproducibility.
> As for retrieval model, we use BGE-m3 embedding model to encode the descriptions of entities, relationships and text chunks into fixed-length vectors. During the execution of multi-step RAG with our HGMem, given a query, the entities, relationships and text chunks with the highest embedding similarity would be retrieved, just as formalised by Equation 3 in Section 3.2. Therefore, as long as the LLMs’ instruction-following capability is strong enough, there is no need for a training process to adapt a native LLM or retrieval model to our HGMem, which is also an advantage of our training-free design.
>
> # References:
>
> [1] Edge et al. From local to global: A graph rag approach to query-focused summarization. arXiv:2404.16130
>
> [2] Guo et al. Lightrag: Simple and fast retrieval-augmented generation. arXiv:2410.05779.
>
> [3] Feng et al. Beyond Graphs: Can Large Language Models Comprehend Hypergraphs?. arXiv:2410.10083.
>
> [4] Xu et al. A-mem: Agentic memory for llm agents. arXiv:2502.12110.

---

> ### Author Response · Authors · 2025-11-26
> **Response to Reviewer h1Xj (Part Two)**
>
> (Cont'd)
>
> # Response to Weakness 3: Reproducibility, Code Release.
>
> At the time of paper submission, although we have completed the experiments of this paper, the codebase is not in a perfectly well-prepared state for final public release. Meanwhile, it is a common practice to fully release the code after acceptance in the community.
>
> Besides, we have provided most of reproduction instructions used in our paper, including necessary experimental settings and important prompts. For the execution of our method, there are basically two stages, namely the offline graph construction and multi-step RAG with HGMem. For the graph construction stage, we just employ the open-source LightRAG toolkit to construct the base graph. For the implementation of multi-step RAG with HGMem, to increase the reproducibility of our paper before final release of full codebase, we further added detailed prompts used for memory evolving and subquery generation in the appendix of our revised paper.
>
> We promised to publicly release our codebase on Github once the codes are well-prepared after the ongoing rebuttal period, so that we will not violate the blindness requirement of ICLR reviewing policy. We hope this will help ease your concern about the reproducibility issue.

---

### Official Review · Reviewer_onmo · 2025-10-29

**Soundness:** 2
**Presentation:** 2
**Contribution:** 3
**Rating:** 2
**Confidence:** 4

**Summary:**

This paper introduces HGMEM, a new memory mechanism for multi-step RAG systems. The core motivation is to address the limitations of existing working memory modules, which often function as passive stores for isolated facts. HGMEM proposes to model memory as a dynamic, evolving hypergraph, where hyperedges represent distinct memory points. Through iterative interactions with external knowledge, this memory structure is progressively refined via three key operations: update, insertion, and merging. The authors argue that this approach enables the formation of high-order correlations among facts, leading to improved reasoning and global understanding in complex, long-context tasks. The method's effectiveness is evaluated on several benchmarks, where it is reported to outperform various traditional and multi-entity RAG baselines.

**Strengths:**

1. The paper  discusses  a timely and relevant limitation in multi-step RAG systems: the static and fragmented nature of existing memory modules. The motivation to enable memory to capture higher-order correlations is well-founded and targets a key bottleneck in complex reasoning tasks.

2. The conceptualization of working memory as a dynamic, evolving hypergraph is an interesting direction. This abstraction provides a potentially powerful framework for representing complex, multi-entity relationships that go beyond the limitations of simpler memory structures.

3. The authors provide an extensive set of experiments on several challenging long-context benchmarks. The reported performance gains,  the results suggesting that an open-source LLM equipped with HGMEM can match or outperform baselines using stronger proprietary models, are notable and merit attention.

**Weaknesses:**

1. The framework's core operations (e.g., merging) are not defined algorithmically but are delegated to a black-box LLM. This introduces uncontrolled non-determinism and makes the method's behavior difficult to formally analyze. The paper does not provide mechanisms or analysis to address the risk of cascading errors stemming from these non-deterministic memory operations.

2. The paper lacks the technical detail required for reproducibility. Key procedures, particularly the 'merging' operation, are described only conceptually. The submission provides no pseudocode or specific prompts that would allow other researchers to implement and verify the method.

3. The method introduces significant computational overhead compared to single-step RAG. The paper fails to provide a quantitative analysis of these costs (e.g., token consumption, latency, number of LLM calls) relative to baselines.

4. The paper fails to cite and discuss highly relevant contemporary work, notably PropRAG (https://arxiv.org/pdf/2504.18070), a highly relevant work that also uses hypergraphs in RAG. Although the two approaches have different goals, the lack of any comparison makes it difficult to situate HGMEM's contribution and assess its novelty against other hypergraph-based methods.

**Questions:**

1. Could the authors provide the specific prompts, pseudo-code, or a detailed algorithmic description for the merging operation?

2. Could the authors provide a quantitative comparison of the computational costs (e.g., tokens, latency) between HGMEM and baselines?

3.  How are contradictions in retrieved information handled? For example, if new evidence conflicts with an existing memory point, does the system have a specific process for resolving this, or does it simply rely on the LLM to implicitly manage the conflict during the next step?

4.  Given that PropRAG also uses hypergraphs in RAG, could the authors elaborate on why it was omitted from the literature review? Please also articulate the unique contributions of HGMEM that are not addressed by prior hypergraph-based RAG methods.

5. The paper's core contribution seems to lie in a conceptual framework and a prompting strategy. This raises a question about the method's robustness and algorithmic nature. Could the authors elaborate on the algorithmic components that are independent of the LLM's specific instruction-following behavior? And how is performance consistency maintained across different base models?

---

> ### Author Response · Authors · 2025-11-26
> **Response to Reviewer onmo (Part One)**
>
> # Response to Weakness 2 and Question 1: Reproducibility,  specific prompts and algorithmic descriptions of merging operation
>
> To address your concern about the reproducibility, we provide detailed prompts for the key procedures of our method in Appendices D and E of our revised paper. Particularly, the prompts for the merging operation is given in Figure 6. We also added the detailed algorithmic description for the merging operation in Section 3.5.
>
> The supplement of our prompts and algorithmic description also supports our clarification to your raised Weakness 1 and Question 5, as explained below.
>
> # Response to Weakness 1, Question 3 and Question 5: Algorithmic nature, robustness and consistency, error resolving
>
> **Clarification about the Algorithmic Nature of Merging Operation**
>
> Based on the provided algorithmic description and prompts (Section 3.5 and Appendix D), it can be seen that the role of LLM is just a functional operator that analyzes current memory and builds high-order correlations among multiple existing memory points to facilitate complex relational modeling. The merging operation itself is intrinsically algorithmic and independent of using what kind of model for relational modeling. Therefore, it is because of the LLM’s strong reasoning capability that we adopt it for obtaining the content of the newly merged memory point, which does not affect the algorithmic nature of our merging operation.
>
> **The Algorithmic Nature and LLM-independent Components of Our Work**
> Although the multi-step RAG execution of our HGMem is indeed driven by LLM-based prompting strategy to a large extent, it cannot deny the overall algorithmic nature of our work.
> Our HGMem also involves the LLM-independent component not specific to LLMs’ instruction-following behaviors, such as adaptive memory-based evidence retrieval (Section 3.4) and merging memory points (Section 3.5, as explained in our preceding response). During the multi-step RAG execution of our HGMem, the LLM is just integrated into the overall framework as a functional operator based on its instruction-following and reasoning capabilities.
> It should be noted that, although prompting-based methods using black-box LLMs would bring non-determinism, they are still widely recognised as algorithmic approaches, which is a prevalent and valid research paradigm applied in various influential researches and applications (e.g. Chain-of-Thought, ReAct).
>
> **Robustness and Consistency across Different Base Models**
>
> In our original experiments (Section 5), we have chosen GPT-4o and Qwen-32B from different model families, as representative closed-source and open-source base models, respectively. In addition, we also add experiments with smaller base models including Llama-8B and Qwen-7B (**please refer to the results given in the Part Four of our response to Reviewer Xaut**). From the results, we can see that our HGMem achieves consistent improvement over other baselines under different base models belonging to different families and of different sizes. These results strongly confirm the robustness and consistency of our HGMem’s superior performance.
>
> **Error/Contradiction Resolving**
>
> First, regarding your concern about cascading error or contradiction during multi-step RAG execution, we would like to respectfully address that the capability of resolving the type of error/contradiction you mentioned is beyond our main targeted research scope.
>
> Secondly, under the targeting task scenarios of our work, the correct resolution of queries does not assume every step of multi-step RAG execution is totally error-free, unlike symbolic reasoning tasks (e.g. math or coding) that requires perfect intermediate procedures. Actually, due to strong reasoning capability of current advanced LLMs, the whole RAG is inherently a noise-tolerant system
>
> Thirdly, the very design of our method—its progressive reasoning structure and dynamic working memory, particularly the update and merging operation, implicitly mitigates the influence of irrelevant or conflicting evidences.
>
> Fourthly, the experimental results have clearly shown that your concerned error/contradiction does not affect the effectiveness of HGMem on our target tasks. If cascading errors or evidence contradiction were a critical bottleneck, our method would not exhibit such robust gains over other baselines across different base models.
>
> Moreover, cascading errors or evidence contradiction is a generic problem existing in all multi-step RAG methods regardless of whether using LLMs, not a particular flaw of our method. We agree that your desired more advanced self-reflection mechanism is actually orthogonal to our current approach, which is a promising future direction to enhance multi-step RAG.

---

> ### Author Response · Authors · 2025-11-26
> **Response to Reviewer onmo (Part Two)**
>
> (Cont'd)
>
> # Response to Weakness 3 and Question 2: Concern about computational overhead
>
> First, it should be noted that, although single-step RAG methods usually have lower costs, they have inherent limitations in dealing with complex problems. Thus, the heavier computational overhead of multi-step RAG compared to single-step RAG is crucial for fully unleashing LLMs’ power, especially for complex relational modeling., which have been demonstrated by many previous studies [1,2,3,4] and real-world applications such as Google DeepResearch system.
>
> This necessitates researchers to achieve a trade-off between cost an performance in practices.
>
> Based on the above reasons, it is of greater importance to compare cost among multi-step RAG systems, which is a common practice taken in many recent researches such as KnowTrace[4] and DeepRAG[5].
>
> For your concern about the quantitative comparison of computational cost with other baselines, we have added the comparison in Appendix C of our revised paper. We also provide detailed qualitative and quantitative cost analysis in our general response to all reviewers. You can refer to the third clarification point in our general response.
>
> # References
> [1] Shao et al. Enhancing retrieval-augmented large language models with iterative retrieval-generation synergy. arXiv:2305.15294.
>
> [2] Trivedi et al. Interleaving retrieval with chain-of-thought reasoning for knowledge-intensive multi-step questions. ACL 2023.
>
> [3] Cheng et al. A survey on knowledge-oriented retrieval-augmented generation[J]. arXiv preprint arXiv:2503.10677.
>
> [4] Li et al. Knowtrace: Bootstrapping iterative retrieval-augmented generation with structured knowledge tracing. KDD 2025.
>
> [5] Guan et al. DeepRAG: Thinking to Retrieve Step by Step for Large Language Models. arXiv preprint arXiv:2502.01142.

---

> ### Author Response · Authors · 2025-11-26
> **Response to Reviewer onmo (Part Three)**
>
> (Cont'd)
>
> # Response to Weakness 4 and Question 4: Novelty against other hypergraph-based RAG methods, comparison with propRAG
>
> ***Before giving our formal responses to your Weakness 4 and Question 4, we hope you can first read our clarification about the difference between short-term working memory and long-term persistent memory in our general response to all reviewers.***
>
> **Unique Contributions of HGMEM**
>
> There are indeed several prior studies that hypergraph-based approaches to improve RAG (e.g. HypergraphRAG, PropRAG). However, most of them belong to the long-term persistent memory, where the hypergraph is built during an offline knowledge indexing stage. (Please refer to our general response).
> In contrast, the hypergraph-based memory in our HGMem is actually a short-term working memory.
> In fact, our HGMem can be orthogonally applied upon those prior RAG methods featuring long-term hypergraph-based memory. In this sense, the uniqueness of HGMem lies in the online usage of hypergraph structure to enhance LLMs’ complex relational modeling for multi-step RAG rather than just utilising offline-constructed hypergraph index.
>
> **About Comparison with PropRAG**
>
> Following the above elaboration, it is clear that although both HGMem and PropRAG incorporate hypergraph structures, their design motivations and application of hypergraph differ fundamentally. The emphasis of PropRAG is to design an LLM-free online search algorithm over offline-constructed hypergraph to discover multi-step reasoning chains while our HGMem focuses on the online evolving (more than searching) of working memory into higher-order structures for complex relational modeling. PropRAG constructs an offline hypergraph and employs hyperedges (Propositions) to expand the retrieval neighborhood, with the primary goal of broadening the set of retrieved candidates. In contrast, HGMem is designed for the online setting: our hypergraph-based working memory evolves dynamically through iterative interaction with external environment, enabling the construction of high-order correlations that directly guide subsequent retrieval decisions. Consequently, despite the superficial similarity in hypergraph usage, the underlying memory representations, operational mechanisms, and underlying objectives of HGMem and PropRAG are substantially distinct.
>
> Even so, we agree that PropRAG is a relevant work to ours in the sense that both adopt hypergraph structure to enhance RAG. So, we add the citation of this work into the related work section of our revised paper. Besides, it should be noted that PropRAG is a very recent work just officially accepted by EMNLP 2025 several weeks before ICLR submission ddl. Considering the huge amount of LLM-related studies posted every day around the globe, it is nearly possible for us to notice all seemingly relevant work and cover them within one paper. This is also the reason why we mainly choose the existing widely compared studies, such as GraphRAG, LightRAG, HippoRAG, in our experiments.
>
> Furthermore, we also compare our HGMem to your PropRAG using Qwen-32B on the tasks used in our paper. Experimental results are given below,
>
> | Method      | NarrativeQA-Narrative (ACC) | Nocha-Book (ACC) | Prelude-English (ACC) |
> |-----------|------------------------------|-------------------|-------------------------|
> | PropRAG | 33.00         | 68.25                        | 51.11             |
> | HGMem     | 51.00         | 70.63                        | 62.22             |
>
> Across all datasets, HGMem consistently achieves remarkbly higher accuracy than PropRAG on these long-context global sensemaking tasks where answers cannot be directly extracted from the original document. In these settings, PropRAG often fails to retrieve sufficiently informative evidence, whereas HGMem’s dynamically evolving working memory enables the formation of high-order correlations that more effectively support complex reasoning.

---

### Official Review · Reviewer_USbV · 2025-11-01

**Soundness:** 2
**Presentation:** 2
**Contribution:** 2
**Rating:** 4
**Confidence:** 4

**Summary:**

Multi-step RAG is widely used to boost LLMs’ performance on tasks requiring global comprehension and intensive reasoning. However, existing memory designs are static passive storage that accumulates isolated facts, ignoring high-order correlations between facts and limiting representational strength. The authors propose HGMEM, a hypergraph-based memory mechanism that transforms memory into a dynamic, expressive structure—hyperedges represent memory units to enable progressive high-order interactions, forming an integrated knowledge structure for subsequent reasoning. Evaluations on global sense-making datasets demonstrate HGMEM consistently enhances multi-step RAG and outperforms strong baselines across tasks.

**Strengths:**

1. This paper identifies an important problem.
3. The proposed method is evaluated on an extensive set of datasets.

**Weaknesses:**

1. The analysis of the connection and difference with existing work is not comprehensive. Firstly, the proposed method is highly related to graph-enhanced iterative RAG, such as SG-Prompt, ERA-CoT, and KnowTrace. More thorough comparison with them is necessary. Secondly, one recent work on long-context understanding, namely CAM (Constructivist Agentic Memory), also aims to tackle the dynamic updating memory and capture the higher-order interactions within memory. It would better to provide more comparisons with this work.
2. The analysis of the proposed method is not clear enough. It would be better to provide an intuitive example.
3. The experiments are not comprehensive. There is a lack of important baselines as described above. Not time complexity and computational cost of the proposed method theoretically and empirically.

**Questions:**

Please refer to the weakenesses.

---

> ### Author Response · Authors · 2025-11-26
> **Response to Reviewer USbV (Part One)**
>
> # Response to Weakness 1: Connection and difference with SG-Prompt, ERA-CoT, KnowTrace and CAM
>
> **Connection**:
>
> SG-Prompt, ERA-CoT and KnowTrace also use graph structure to enhance model’s understanding and reasoning.
> KnowTrace also employs structured multi-step reasoning strategy to overcome the limitation of purely unstructured information accumulation, where the knowledge graph serves as a working memory that continuously expands and supports knowledge exploration and backtracking.
>
> **Differences**:
>
> To clarify the difference of our HGMem with your mentioned work, we provide more detailed comparisons from the following aspects:
>
> (1) Inherently Different Task Characteristics:
>
> The target tasks of your mentioned methods are inherently different from those our HGMem aims at. One is just for fact-oriented multi-hop problems that can be well-resolved through accurate retrieval, while the other is for global sense-making tasks that require complex relational modeling. You can refer to the first point of our clarification in general response to all reviewers for more detailed explanation. Such inherent task difference fundamentally motivates our design to adopt a hypergraph-based memory structure to support complex relational modelling (high-order correlation) for improving multi-step RAG, as elaborated in the below paragraph regarding Methodological Distinction.
>
> (2) Methodological Distinction
>
> Your mentioned work basically use their knowledge graphs as memory modules to store primitive low-order evidences accumulated during multi-step graph-enhanced RAG execution. However, the graph structures of the memory modules in these systems do not effectively support modeling high-order correlations among multiple entities/relationships over long context. Each edge in their graphs can only describe at most binary relationships between two nodes, inevitably restricting the capability and expressiveness of their memory modules to support complex relational modeling. By contrast, due to the intrinsically high-order nature of hypergraph structure, our HGMem enables its memory to evolve into more expressive forms capable of modeling high-order n-ary (n>2) relations. This actually support complex relational modeling and facilitate deep understanding to fully unleash the reasoning capability of LLMs, especially suitable for dealing with global sense-making tasks we are aiming at.
>
> To solidly validate this claim, we also conduct supplementary experiments to observe the performance of your mentioned work on the long-context understanding tasks used in our paper. Specifically, we compare our HGMem with KnowTrace, the strongest one among your mentioned methods. The results show that our target tasks really poses great challenges to these existing works. You can refer to the results of this supplementary experiment in our response to your raised weakness 3.
>
> **Comparison with CAM**
>
> CAM proposes to assimilate and accommodate the information extracted from input texts within a hierarchical graph. Although it also dynamically update memory and capture interactions within memory, the memory mechanism in CAM is actually a long-term memory instead of working memory. For the distinction between working memory and long-term memory, you can also refer to the second point of our clarification in general response to all reviewers. Same as other studies of long-term memory, CAM builds its hierarchical graph in an offline graph construction stage, not specific to any user query. In fact, this means that CAM is naturally orthogonal to our HGMem, because our hypergraph-based working memory can be directly applied upon CAM.
>
> Besides, at each layer of the hierarchical graph of CAM, due to the nature of its graph structure, each edge can only embody a binary relationship two nodes, unlike the hyperedges in a hypergraph flexibly carrying high-order correlation among multiple vertices.  The effectiveness of such flexible expressiveness has been validated by many prior hypergraph-based researches [1,2].
>
> In our revised version, following your suggestion, we added the citations of ERA-CoT, KnowTrace and CAM in related work section, and provided comparison with our HGMem.
>
> # References
> [1] Luo et al. HyperGraphRAG: Retrieval-Augmented Generation via Hypergraph-Structured Knowledge Representation. arXiv:2503.21322.
>
> [2] Hu et al. Cog-RAG: Cognitive-Inspired Dual-Hypergraph with Theme Alignment Retrieval-Augmented Generation. arXiv:2511.13201.

---

> ### Author Response · Authors · 2025-11-26
> **Response to Reviewer USbV (Part Two)**
>
> (Cont’d)
>
> # Response to Weakness 2: The analysis of the proposed method is not clear enough. It would be better to provide an intuitive example.
>
> ***To address your concern, in the revised version of our paper, we have added a case study section in the Appendix G, which presents two representative cases highlighting HGMem’s distinct reasoning advantages over LightRAG from the perspective of forming high-order correlations and the strategy of adaptive memory-based evidence retrieval during memory evolving.***
>
> The first case is from NarrativeQA dataset, where the question requires inferring the underlying cause of Xodar’s enslavement—a relation not explicitly stated in the original text. LightRAG just makes incorrect surface-level predictions based on the retrieved content. While DeepRAG stores the knowledge in the memory, it does not form high-order correlation and fails to predict correctly. In contrast, HGMem progressively evolves its memory and establishes high-order correlations from primitive evidences accumulated from past interactions, uncovering that Xodar’s punishment originates from his defeat by Carter.
> The second case is from Nocha dataset, where the query mixes factual and misleading details. The LLM raises a subquery about the source of the name ‘White Sands’. LightRAG mistakenly recognizes that the name ‘White Sands’ was given by Anne and DeepRAG doesn’t qualify the correctness of ‘White Sands’. By contrast, based on the strategy of local investigation for evidence retrieval, our HGMem conducts in-depth inspection about the related memory point (Point 1) in current memory and verifies that there is no clear evidence showing the name was given by Anne.
>
> Together, these examples show that HGMem facilitates a deeper and more accurate contextual understanding from the following two perspectives:
>
> (1) HGMem dynamically evolves its hypergraph-based memory to form high-order correlations beyond primitive evidence accumulated during multi-step RAG.
>
> (2) The adaptive evidence retrieval strategy effectively guides the execution of multi-step RAG.
>
>
> # Response to Weakness 3: Lack of baselines as described above. No cost analysis of the proposed method.
>
> To address your concern about important baselines, we further compare our HGMem with  KnowTrace using Qwen2.5-32B on the long-context understanding tasks in our paper, namely long-context understanding tasks. We choose to compare KnowTrace because
>
> (1): KnowTrace exhibits higher performance than SG-Prompt and ERA-CoT according to the results reported in their papers.
>
> (2): As explained in our response to your raised weakness 1, CAM is orthogonal to our proposed HGMem.
>
> The comparison results are as follows.
>
> | Method      | NarrativeQA-Narrative (ACC) | Nocha-Book (ACC) | Prelude-English (ACC) |
> |-----------|------------------------------|-------------------|-------------------------|
> | KnowTrace | 44.00         | 69.04                        | 44.44             |
> | HGMem     | 51.00         | 70.63                        | 62.22             |
>
>
> We can see that our HGMem exhibits significantly better performance over KnowTrace, demonstrating the superiority of our proposed method.
>
> Besides, it is worthy to mention that DeepRAG, as one of the compared baseline methods in our original paper, outperforms SG-Prompt, ERA-CoT and KnowTrace under matched model sizes and evaluation settings (please see their reported results on Llama3-8B on HotpotQA and 2WikiMultihopQA) according to the results from both Deeprag and KnowTrace papers. This further implies that DeepRAG has already been a representative baseline competitive enough to validate the effectiveness of our HGMem over your mentioned important baselines.
>
> For the time complexity and computational cost of the proposed method, please refer to the third point of our clarification in our general response to all reviewers.

---

### Author Response · Authors · 2025-11-26
**General Response to All Reviewers (Part Three)**

(Cont'd)

**3. LLM Cost Analysis of our HGMem. (Reviewers onmo and Xaut)**

As several reviewers commonly raised concerns about the cost (token consumption and latency) of our proposed HGMem, we would like to provide our analysis from both qualitative and quantitative perspectives. Meanwhile, it should be noted that the cost and latency of different RAG systems is greatly affected by their own configurations and implementations, e.g. prompting strategy, number of retrieved passages, etc.
Therefore, in practice, the primary goal should be achieving a trade-off between cost and performance.

***Qualitative Analysis***

The LLM cost of our HGMem is basically composed of the following operations: (1) update\&insert memory points; (2) merging memory points; (3) subquery generation; (4) final response.
Theoretically, all memory-enhanced RAG methods need operations (1)(4), regardless of single-step or multi-step RAG systems. Therefore, (2) and (3) are the only additional LLM-based operations introduced in our HGMem during multi-step RAG execution, which have been proven to be crucial for final performance improvement according to our experimental results (Table 3).
In fact, the operation (2) is essentially a kind of LLM-based structural memory organising operation that aims at refining memory storage. This type of structural memory organising operation has been widely implemented in many existing RAG systems in various forms, such as community summarisation (GraphRAG, CAM, A-Mem, etc), entity-relation analysis (ERA-CoT).
Meanwhile, the operation (3) is to decompose the target query into subqueries for facilitating complex reasoning, which is also implemented by many multi-step RAG systems, such as DeepRAG, ComoRAG.
In this sense, compared to most existing studies, our HGMem does not heavily introduce additional LLM-based operations.

***Quantitative Analysis***
To show the computational efficiency of our approach, we provide a quantitative comparison about the online multi-step RAG execution cost across HGMem and the relevant baselines (DeepRAG and ComoRAG). We emphasize that online cost is the appropriate metric for comparison, as the offline graph construction only builds a query-agnostic indexing structure and therefore lies outside the scope of our analysis.
The table below reports the average count of tokens as well as inference latency per query. As shown, HGMem exhibits token consumption and latency that are roughly comparable to those of DeepRAG and ComoRAG, while achieving consistently stronger performance across all datasets.
Regarding the concern that addtional merging operation may introduce extra computational cost, our ablation (w/o merging) indicates that the merging step only introduces minor increases in both computation and latency. This is as expected since the LLM is just responsible for generating the textual description of new higher-order hyperedges. The actual topological update on the hypergraph is supported by the hyperdb package without generating a substantial number of additional tokens. That is to say, forming high-order correlations enabled by merging provides notable performance gains while incurring only lightweight computational overhead.

| **Method**            | **NarrativeQA Avg-Token** | **NarrativeQA Avg-Time** | **Nocha Avg-Token** | **Nocha Avg-Time** | **Prelude Avg-Token** | **Prelude Avg-Time** |
|-----------------------|---------------------------|---------------------------|----------------------|---------------------|------------------------|-----------------------|
| **HGMem**             | 4436.43                   | 15.84                     | 5252.73              | 18.76               | 5421.74                | 19.36                 |
| *w/o. Merging*        | 4154.02                   | 14.84                     | 4750.32              | 16.97               | 4897.81                | 17.49                 |
| **DeepRAG**           | 3904.18                   | 13.94                     | 4724.07              | 16.87               | 4514.66                | 16.12                 |
| **ComoRAG**           | 5083.26                   | 18.15                     | 5503.98              | 19.66               | 7827.56                | 27.96                 |

---

### Author Response · Authors · 2025-11-26
**General Response to All Reviewers (Part Two)**

(Cont'd)

Moreover, we would like to clarify several important points about some common concerns raised by reviewers. We hope that this clarification would help you understand our work better and reinforce our contribution.

**1. The targeted tasks of our proposed method are global sense-making tasks that require complex relational modeling beyond mere primitive evidence retrieval.**

As stated in the paper introduction, the primary goal of our work is to address the limitation and challenge of current multi-step RAG methods on long-context understanding tasks that involve complex relational modeling, especially global sense-making questions whose answers cannot be directly derived by accurately retrieving primitive evidences.
Although many existing RAG approaches claim to be dealing with so-called long-context understanding task, most of them still mainly focus on intrinsically fact-oriented multi-hop problems that can be well-resolved as long as necessary local evidences are accurately retrieved (e.g. HotpotQA). The typical characteristic of such problem is that its expected reasoning trajectories can be clearly decomposed into subquestions featuring shallow evidence retrieval. Given the question “Which magazine was started first Arthr's Magazine or First for Women?” (from HotpotQA), the question can be clearly decomposed into “subq1: When was Arthr's Magazine Started?” and “subq2: When was Women Started”. Every subquestion is actually still a fact-oriented problem that just necessitates accurate local evidence retrieval. In a sense, such problems do not really require LLMs’ long-context understanding capability in an RAG setting.

In contrast, our paper primarily targets at global sense-making questions whose answers cannot be derived by merely retrieving accurate primitive evidences but further requires complex relational modelling over long contexts (even the entire document). As an illustrative example, let’s look at the case given in the Appendix G - “Why is Xodar given to Carter as a slave?”. To resolve this question, even though ‘Xodar’ and ‘Carter’ are apparently the target entities to investigate, it requires LLMs to gain a deep understanding about relevant story plots. To ultimately arrive at the correct answer, the LLM should comprehend complex relationships among multiple related characters and events over long contexts. Such inherent task difference fundamentally motivates our design to adopt a hypergraph-based memory structure to support complex relational modelling (high-order correlation) for improving multi-step RAG.

**2. The hypergraph-based memory in HGMem is a short-term working memory rather than long-term persistent memory.**

After reading all comments, we find it necessary to indicate the fundamental difference between short-term working memory and long-term persistent memory, because reviewers seem not to distinguish between the two lines of studies.

Following the taxonomy used in [1,2],

Working memory is usually query-specific, online built and dynamically maintained during the process of resolving user queries, temporarily storing accumulated information around a focal problem. For instance, DeepRAG, ComoRAG, ERA-CoT and KnowTrace belongs to working memory mechanism.

By contrast, long-term persistent memory mostly resorts to storing information about data sources using external storage (e.g. database), which is typically built through offline indexing. Although its formation process also involves dynamic updates, it is still intrinsically a static storage once being built, and thus not specific to any user query. Subsequently, RAG interactions are executed over this static long-term memory using indexing operations (so-called memory retrieval). Particularly, GraphRAG, LightRAG, HippoRAG, CAM, PropRAG, A-Mem, THEANINE, Zep belong to this line of long-term memory mechanism.

According to the above criteria, the hypergraph-based memory in our HGMem is obviously a short-term working memory rather than long-term memory. Theoretically, any working memory mechanism can be orthogonally applied upon long-term memory.

## References
[1] Wu et al. From human memory to ai memory: A survey on memory mechanisms in the era of llms. arXiv preprint arXiv:2504.15965.

[2] Du et al. Rethinking Memory in AI: Taxonomy, Operations, Topics, and Future Directions. arXiv preprint arXiv:2505.00675.

---

### Author Response · Authors · 2025-11-26
**General Response to All Reviewers (Part One)**

Dear All Reviewers,

First and foremost, we are greatly grateful to the time and effort you put into reviewing our paper.

**As a follow-up to all your feedback, we have highlighted all revisions of our revised version in yellow. We revised our paper in the following aspects**:

(1) To enhance the reproducibility, we provide the detailed prompts of the key procedures in our method (See Figures 5,6,7,8 in Appendix), which would allow other researchers to implement and reproduce. (Reviewers onmo and h1Xj)

(2) We add a case study section (Appendix G) to provide intuitive examples for analysing our proposed method, highlighting our HGMem’s distinct advantages over representative prior work. (Reviewers USbV and h1Xj)

(3)We provide a toy example (Appendix H) to clearly illustrate the detailed process of multi-step RAG executions in our HGMem. (Reviewer h1Xj)

(4) We refine the writing of methodology (Section 3) to improve the algorithmic description of our proposed method, especially adding necessary symbolic definitions in sections 3.3 and 3.4. (Reviewers onmo and h1Xj)

(5) We refine the introduction section to state the objective and task scope of our work more clearly. (Reviewer h1Xj)

(6) We refine the related work (Section 2) to provide more comparisons with your mentioned work that are relevant to our method including ERA-CoT, KnowTrace, CAM, PropRAG. (Reviewers USbV and onmo)

(7) We provide a cost analysis section in the Appendix C of our revised paper. (Reviewers USbV, onmo and h1Xj)

---

### Meta-Review · Area_Chair_8uaP · 2026-01-08

**Summary:**

The paper proposes HGMEM, a hypergraph-based working memory mechanism for multi-step RAG. The authors address the limitations of static memory by modeling working memory as a dynamic hypergraph where hyperedges (memory points) represent high-order correlations among entities. The framework is training-free and relies on an LLM-driven pipeline to update, insert, and merge these memory points during inference. The method is evaluated on long-context global sense-making tasks (e.g., NarrativeQA) and is shown to outperform baselines such as GraphRAG and LightRAG.

**Reviewer Concerns:**

Addressed Concerns:
1. Missing Baselines: Reviewers USbV, onmo, and Xaut noted the absence of relevant comparisons (e.g., KnowTrace, PropRAG, A-Mem). The authors conducted extensive new experiments during the rebuttal, demonstrating HGMEM's empirical superiority over these baselines.
2. Reproducibility: Reviewers onmo and h1Xj criticized the lack of prompts and algorithmic details for the merging operation. The authors updated the appendix with specific prompts and symbolic definitions to improve reproducibility.
3. Computational Cost: The authors provided a cost analysis (Appendix C) to address concerns from Reviewers USbV and onmo, showing that the method's latency is comparable to other multi-step RAG systems like DeepRAG.

Outstanding Concerns:
1. Engineering vs. Principled Research: While the authors have polished the system to achieve good results, the core contribution remains a sophisticated exercise in pipeline engineering and prompt tuning rather than a fundamental advancement in representation learning. As Reviewer Xaut noted, HGMEM does not offer a new learning algorithm or principled framework but rather a heuristic application of existing LLMs.
2. Black-Box Dependency: As highlighted by Reviewer onmo, the critical merging and updating operations are not algorithmic in a traditional sense but are delegated to the black-box reasoning of the LLM via natural language prompts. This introduces non-determinism and limits the scientific insight into how the high-order correlations are actually modeled.

**Reviewer Scores:**

no changes

---

### Decision · Program_Chairs · 2026-01-26

Reject